# Peritoneal Metastasis: A Dilemma and Challenge in the Treatment of Metastatic Colorectal Cancer

**DOI:** 10.3390/cancers15235641

**Published:** 2023-11-29

**Authors:** Wei Xia, Yiting Geng, Wenwei Hu

**Affiliations:** 1Department of Oncology, The Third Affiliated Hospital of Soochow University, 185 Juqian Street, Changzhou 213003, China; wxiaxw9938@stu.suda.edu.cn; 2Jiangsu Engineering Research Center for Tumor Immunotherapy, The Third Affiliated Hospital of Soochow University, Changzhou 213003, China

**Keywords:** colorectal cancer (CRC), peritoneal metastasis (PM), intraperitoneal free cancer cells (IFCCs), cytoreductive surgery (CRS), intraperitoneal chemotherapy (IPC)

## Abstract

**Simple Summary:**

The peritoneum, a common metastatic site of colorectal cancer (CRC), has a high incidence and poor prognosis that makes it difficult to diagnose early. Peritoneal metastasis (PM) depends on the synergistic action of multiple molecules and the regulation of various components of the tumor microenvironment. A multidisciplinary combination approach is still recommended for treating the disease currently. Cytoreductive surgery (CRS) combined with intraperitoneal chemotherapy (IPC) may benefit patients with CRC-PM, but further clinical trials and higher-level evidence-based medical evidence are needed.

**Abstract:**

Peritoneal metastasis (PM) is a common mode of distant metastasis in colorectal cancer (CRC) and has a poorer prognosis compared to other metastatic sites. The formation of PM foci depends on the synergistic effect of multiple molecules and the modulation of various components of the tumor microenvironment. The current treatment of CRC-PM is based on systemic chemotherapy. However, recent developments in local therapeutic modalities, such as cytoreductive surgery (CRS) and intraperitoneal chemotherapy (IPC), have improved the survival of these patients. This article reviews the research progress on the mechanism, characteristics, diagnosis, and treatment strategies of CRC-PM, and discusses the current challenges, so as to deepen the understanding of CRC-PM among clinicians.

## 1. Background

### 1.1. Epidemiological

Colorectal cancer (CRC) is one of the most common malignancies, with about 1,881,000 new cases and 916,000 deaths from the disease worldwide every year [1], posing a serious threat to human health. Distant metastases are the leading cause of death due to treatment failure. The most likely sites for CRC to metastasize are the liver and lung [2], and the peritoneum is also a common site for CRC metastasis. Compared with the liver and lung, peritoneal metastasis (PM) in CRC cases usually means that patients lose the possibility of conversion therapy and have a worse prognosis. It has been reported that PM is discovered in approximately 13% of CRC cases, with 2% involving isolated PM and the remaining 11% being peritoneal and combined with other organs [3]. Patients who suffer from PM develop a higher proportion of BRAF mutations and worse survival. Due to the lack of effective treatment, the average survival of such patients was shorter than patients with liver-isolated metastasis (16.3 months (13.5–18.8) vs. 19.1 months (18.3–19.8)). Furthermore, PM combined with ≥2 sites of metastases has a 1.4-fold increased risk of death compared with PM alone [3]. 

### 1.2. Occurrence of PM

The “seed and soil” idea states that tumor cells interact with milky spots, the peritoneal surface features, to create a milieu for tumor cell colonization and dissemination [4]. Based on the generation time of the primary lesion and PM, it can be divided into synchronous PM and metachronous PM. The former is defined as PM that is detected at the time of diagnosis of CRC or observed within 30 days after primary tumor resection; the latter is defined as PM observed 30 days after radical colorectal resection [5]. The spilling of cancer cells during tumor growth causes synchronous PM. Besides the intraperitoneal free cancer cells (IFCCs) caused by tumor itself, medical variables during surgery can shed cancer cells into the peritoneal cavity, causing metachronous PM.

### 1.3. Diagnosis and Treatment Development

PM cancer was previously considered the final stage of disease development, with a median survival of 3.1 months [6], and little hope of cure or benefit from surgery. Sugarbaker’s 1980s treatment strategy combined cytoreductive surgery (CRS) with hyperthermic intraperitoneal chemotherapy (HIPEC), transforming peritoneal carcinoma (PC) from an incurable to a treatable and even curable disease [7]. In 2003, Verwaal et al. found that this treatment mode can significantly improve patient prognosis [8]. Over the past 20 years, increasing evidence has supported the inclusion of CRC/HIPEC in the treatment of PC. This local therapeutic method and new multidisciplinary strategies have transformed CRC-PM prognoses.

The heterogeneity of CRC-PM in terms of pathogenesis, clinical manifestations, treatment methods, and adverse effects on the survival of patients makes it a special metastasis type M1c that is different from other metastases in the eighth edition of the AJCC stage, which requires clinicians to focus on different treatments in diagnosis and therapy. This article reviews the mechanism and process of occurrence, molecular biological characteristics, clinical manifestations, and diagnostic methods of CRC-PM and summarizes the existing therapeutic means, as well as representative clinical studies, to provide reference for the prevention and control of CRC-PM.

## 2. Pathophysiological Process and Molecular Biology Characterization of PM 

### 2.1. Pathophysiological Process 

The peritoneum comprises the mesothelial cell layer, connective tissue, vascellum, and lymphatic tissue. CRC-PM can be achieved via hematologic, lymph node, or direct implantation. PM performs as a series of complex pathophysiological processes, the so-called “PM cascade”, involving multiple cellular, molecular, and genetic alterations, which create a special peritoneal microenvironment conducive to the CRC-PM. 

The decrease in the expression or functional levels of intercellular adhesion molecule (ICAM), such as E-cadherin, causes CRC cells to disperse from the primary tumor into the peritoneal cavity, either singly or in clusters, and become IFCCs [9]. Epithelial–mesenchymal transition (EMT) [10] plays an important role in the acquisition of stem cell properties and motility/migration phenotypes by cancer cells. Cellular motility is augmented, extracellular matrix (ECM) and cell–cell adhesion are compromised, apic-base polarity is eliminated, and the cytoskeletal structure is reorganized throughout EMT. Under the contraction of actin contraction, each of these aggressive phenotypes promotes cancer cells to separate from the cancer nest, migrate, and invade [11]. The loss of epithelial characteristics in aggressive frontier tumor cells and the adoption of mesenchymal-like phenotypes in CRC leads to enhanced aggressiveness [12]. These modifications cause tumor cells to resemble normal cells during embryonic development, allowing tumor cells to adapt to shifting microenvironments and successfully spread.

The spread of IFCCs in the peritoneal cavity is not a random process. Under the influence of respiration, gravity, and intestinal peristalsis, intra-abdominal pressure changes and leads certain areas of the peritoneal cavity, such as the subdiaphragmatic region, the lesser sac, mesentery, diaphragm, and paracolic sulcus, to be at an increased risk of metastasis [13]. On the one hand, IFCCs secrete various pro-inflammatory cytokines, such as tumor necrosis factor (TNF)-α, interleukin (IL)-1β, IL-6, and interferon (IFN)-γ, which promote the high expression of ICAM, platelet endothelial cell adhesion molecule (PECAM), vascular adhesion molecule (VCAM) as well as other immunoglobulin superfamily cell adhesion molecules (IgCAMs) via human peritoneal mesothelial cells (HPMCs), and induce mesothelial cell contraction and roundness for the exposure of the basement membrane. On the other hand, the adhesion of IFCCs to HPMCs is mediated by the specific binding of CD43 (sialophorin) on the cell surface to the IgCAMs widely expressed by HPMCs above [14]. Thereafter, HPMCs produce lysophosphatidic acid, which can further promote cancer cell adhesion and form “positive feedback”. In addition, IFCCs express CD44 molecules, a cell surface glycoprotein that interacts with hyaluronic acid secreted by mesothelial cells [15], resulting in tumor cell mesothelial adhesion. A small number of tumor cells invade the mesothelium and induce HPMCs apoptosis through FasL/Fas [16], disrupting peritoneal continuity and invading the subperitoneal space.

Matrix metalloproteinases (MMPs) play a significant role in CRC metastasis to the peritoneum. When tumor cells invade the subperitoneal space, surrounding cells like mesothelium, fibroblasts, inflammatory cells, and macrophages secrete MMPs, leading to the degradation of ECM [17,18]. In addition, the urokinase-type plasminogen activation system also disrupts the peritoneal barrier by activating pro-MMPs and degrading ECMs [19]. Subsequently, tumor cells along with stromal cells generate various growth factors represented by epidermal growth factor receptor (EGFR) and insulin-like growth factor 1 (IGF-1) via autocrine and paracrine pathways to stimulate the continuous proliferation of cancer cells [20]. Under a hypoxic environment, the expression of hypoxia-inducible factor 1α (HIF-1α) is significantly elevated in PM, further activating the transcription of angiogenesis-related genes including the vascular endothelial growth factor (VEGF) family [21,22], promoting the development of new microvessels and lymphatic vessels to obtain nutrients and oxygen, and finally leading to the formation of lymphatic metastasis and cancerous ascites (Figure 1).

Tumor-associated fibroblasts (CAFs) are the most prevalent cell type in the tumor microenvironment (TME) and the hub of tumor mesenchyme cell communication. Lipid metabolism reprogramming also occurs in CAFs, whose secreted fatty acids and phospholipids are taken up by CRC cells and can promote CRC cell migration [23]. CAFs can upregulate fatty acid oxidation rate-limiting enzyme CPT1A to actively oxidize fatty acids, thereby promoting CRC-PM [24]. In addition, they can promote the migration of CRC-PM through the upregulation of unsaturated acyl chains in phosphatidylcholine to increase cell membrane fluidity, thereby increasing CRC cell migration and intraperitoneal spreading. Treatment with sodium palmitate (C16:0) reduced CAFs-induced changes in cell membrane fluidity and inhibited tumor growth and intraperitoneal spreading [25], uncovering new opportunities for the future treatment of CRC-PM.

PM is caused by a multifactorial, multi-stage, and multi-gene interaction between tumor biology, peritoneal biology, and biological cytokines. Identifying and exploring the mechanism and theory underlying CRC-PM will help clinicians prevent and treat it better.

### 2.2. Molecular Biology Characterization

Although the molecular features of CRC-PM are not yet well understood enough to influence treatment decisions, a better understanding of molecular biology, including genomics, transcriptomics, and proteomics, can help quantify the risk of PM in individual patients after initial radical surgery and may be useful in predicting the benefits of specific drugs, advancing effective drug development and improving patient prognosis.

#### 2.2.1. High-Frequency Mutations in PM

Precision medicine is based on individual bioinformatics using high-throughput technologies such as next-generation sequencing (NGS). In a genome-wide analysis by the Cancer Genome Atlas Network in 2012, mutations with high frequency in CRC primary foci included WNT (APC, CTNNB1, SOX9, TCF7L2, DKK, AXIN2, FBXW7, ARID1A, and FAM123B), PI3K, RAS-MAPK (IGF2, IRS2, PIK3R1, PIK3CA, PTEN, KRAS, NRAS, BRAF, ERBB2, and ERBB3), transforming growth factor-β (TGF-β) (TGFBR1, TGFBR2, ACVR2A, ACVR1B, Smad2, SMAD3, and Smad4), and p53 (53 and ATM) [26]. Although CRC-PM differed in mutation frequency, the mutation pattern was similar to primary CRC: TP53 (median 54%, 33–75%), KRAS (45%, 20–58%), APC (44%, 31–57%), Smad4 (22%, 15–29%), BRAF (15%, 6–36%), and PIK3CA (13%, 9–14%) [26]. 

There are some discrepancies in the current studies on KRAS and BRAF in CRC-PM. Some studies have suggested that in patients with CRC-PM, the rates of KRAS and BRAF mutations are similar to those of primary CRC [26,27,28,29,30], but others have found that PM has a higher rate of BRAF mutations (25–36%) than primary CRC (5–10%) [31,32]. In some studies, BRAF mutations are a negative prognostic marker for patients with CRS/HIPEC [28,29,33]. However, it has also been shown that BRAF mutation status does not influence survival [34]. There are similar differences in KRAS. KRAS and KRAS codon 12 mutation was significantly associated with PM [35]. RAS mutations also affected survival after CRS/HIPEC [27,36] and their mutation status could be used as a marker for predicting peritoneal recurrence [37], while other studies showed the opposite [28,29,34]. 

BRAF-mutant tumors in metastatic colorectal cancer (mCRC) have a greater incidence of PM and are substantially linked to microsatellite instability (MSI) [38]. Compared to individuals with low-frequency MSI (MSI-L)/microsatellite stability (MSS) CRC, those with high-frequency MSI (MSI-H) CRC had a higher chance of developing PM [39]. The expression of PD-1, tumor mutation load, and MSI-H were lower in CRC-PM than in primary tumors [30], suggesting that CRC-PM has a unique molecular expression profile. Microsatellite (MS) status is also a stratification factor for CRC-PM patients undergoing CRS/HIPEC, with MSI having a better prognosis [34,40], which may be one of the reasons for the differences between KRAS and BRAF in different studies.

PIK3CA mutation was a protective factor against PM, and it was not only significantly associated with a lower chance of PM at the initial diagnosis of mCRC (OR 0.10, 95% CI 0.01–0.79) but it also dramatically reduced the risk of PM at any time thereafter (HR 0.31, 95% CI 0.11–0.86) [41]. But, a recent study indicated that PI3K pathway modifications after CRS in CRC-PM patients were related to a drop in recurrence-free survival (RFS, 5 vs. 13 months), it represented a poor prognostic indicator, and it was a new molecular subtype of PM associated with early recurrence [42].

In primary CRC, increased mutation burden in the tumor suppressor gene FBXW7 is associated with a lack of distant metastasis and increased expression of T cell proliferation and antigen presentation [43]. In contrast, FBXW7 mutation burden is lower in PM [30], which means it may play a role in the CRC-PM.

Among the genes repeatedly mutated by NGS (TP53, APC, Smad4, PIK3CA, and FBXW7), only the APC mutation (36.8%) was lower than the general level of mCRC, which negatively correlated with CRC-PM and suggested better survival. However, it is unclear whether this is a CRC-PM-specific alteration or the result of a higher proportion of right hemi-primary foci [44]. The role of APC mutations in the occurrence and progression of CRC-PM needs to be further explored in additional studies. 

CRC-PM showed significant molecular heterogeneity. In the same individual, key driver genes such as KRAS, APC, and TP53 mutations were homogeneous across samples from CRC-PM patients, special AT-rich sequence-binding protein 2 (SATB2) lacked expression in most cases, while less common mutations such as RNA-binding motif protein 3 (RBM3) showed significant heterogeneity in different samples from the same patients. Similarly, copy number variation (CNV) is heterogeneous both within and between patients [45]. There are a few studies on CNV in CRC-PM; small sample studies have reported increased expression of 5P and 12P genes in PM compared to primary CRC foci or liver metastasis (LM) in the past [46,47].

Due to its varying molecular expression patterns from the initial lesion, more research is needed to determine the genome/epigenomic profile of CRC-PM, which will aid in patient prognosis and treatment.

#### 2.2.2. Consensus Molecular Subtypes (CMS)

CMS4 might be the major phenotype of CRC-PM. CMS staging has a clear biological interpretation and is currently the most reliable classification system for CRC. It is also the basis for future clinical stratification and subtype-based targeted interventions, as shown in Figure 2 [48,49].

Among the four types of CMS, CMS4 has the largest association with CRC-PM, which is clearly attributed to its molecular properties: enhanced tumor cell invasiveness due to TGF-β activation mediating EMT, as well as abundant and active angiogenesis, all of which constitute favorable conditions for PM. This has been validated in sequencing analyses based on primary CRC and PM tissue samples. 

In the study by Ubink et al., CRC-PM was not only associated with poor histopathological features, such as a higher proportion of mesenchymal components in primary tumors and metastases, a low degree of differentiation in primary tumors, and high levels of tumor budding, but also a significantly higher percentage of CMS4 in primary tumors with PM (60% vs. 23%, *p* = 0.002) and in PM, CMS4 enrichment was even higher (21/28, 75%), with at least one CMS4-positive tumor site was present in 15 out of the 16 patients with paired tumors [31]. The higher proportion of mesenchymal, and stronger propensity for hematogenous metastasis makes such tumors appear to have some of the characteristics of sarcomas and thus may benefit from therapies targeting the tumor mesenchymal, such as anti-vascular therapy. It is important to note that there is considerable heterogeneity in tumor CMS4 status within patients, with inconsistent CMS4 classification between primary tumors and PM in 50% of patients [31], suggesting that molecular phenotypes and signaling change markedly during CRC-PM and the CMS typing remains a limitation for the identification of molecular biological properties of CRC-PM. 

It was also discovered in later research that nearly all tissue biological samples from patients with CRC-PM’s original tumors and PM foci were CMS4 categorized, but the expression of CMS4 recognition genes varied greatly between groups and was significantly higher in PM than in primary tumors, thereby distinguishing a population group with a poor prognosis. Moreover, the CMS4 status of CRC-PM and the high reductive capacity due to increased glutathione synthesis also led to oxaliplatin (OXL) resistance [50]. In addition, this study demonstrated that 15 Hallmark pathways, including TGF-β signaling, angiogenesis, complement activation, and EMT, were expressed at significantly higher levels in PM than in their corresponding primary tumors, while WNT signaling and MYC target gene expression were significantly reduced. The presence of immature dendritic cells (DCs), monocytes, macrophages, and natural killer (NK) cells was significantly more characteristic in PM compared with their primary tumors [48,51]. These are all characteristics of CMS4 CRC and reflect the more extreme mesenchymal phenotype of CRC-PM. ZEB1 is a major regulator of EMT and can be used to identify CMS4 CRC in situ [52]. So the highly mesenchymal phenotype of CRC-PM may be associated with their high expression of ZEB1.

However, there are different views. In Barriuso’s study, CMS2 was the subtype that comprised the highest proportion and had prognostic predictive value [53]. The difference in the results shown may be attributed to the fact that both studies had small sample sizes and used assays built on different platforms.

More transcriptomic or proteomic studies should be encouraged to further clarify the complex downstream signaling pathways and the potential differences between CRC-PM and other metastatic types. This is necessary because the above findings are only a preliminary investigation of the molecular biology of CRC-PM, and most of the studies had small sample sizes. Furthermore, the samples collected were from a subset of patients with indications for surgery, which introduces selection bias. Many subjects also had distant metastases from other sites or included solid tumors of completely different pathological types, such as peritoneal pseudomyxoma and appendiceal malignancies. Additionally, the experiment’s selection of microarray and sequencing panels exhibited significant variability, with all these factors leading to divergent results among studies. The lack of an in-depth exploration of the underlying mechanisms behind the appearances further complicates the understanding of CRC-PM. To address these limitations, it is crucial to establish large, representative patient cohorts and standardized sample collection, processing, and analysis. Such efforts would not only improve the prognosis of these refractory patients but also guide individualized therapy.

## 3. Characterization of Tumor Immune Microenvironment

The peritoneum comprises a multilayered structure of a polysaccharide envelope, mesothelium, basal lamina, submesothelial connective tissue, and elastic lamina. Its main role is to regulate homeostasis in the peritoneal cavity; in addition, the peritoneum plays an important role in developing antigen presentation, inflammatory response, fibrosis and fibrinolysis, tissue repair, and tumor metastasis [54]. The immunoreactive cells of multiple components are present in the peritoneal cavity, including mononuclear macrophages (42%), T lymphocytes (40%), NK cells (10%), and DCs (5%) [55] (Figure 3), with the majority of CD4^+^ and CD8^+^ T cells being the CD45RO^+^ phenotype, representing differentiated memory and effector T cells that mediate specific immune responses against pathogens such as bacteria and viruses as well as cancer cells.

### 3.1. Differences between PM and Primary Foci

According to the available evidence, CRC primary lesions and accompanying PM differ greatly in the immunological environment. The proliferation rate of tumor cells in PM was significantly lower compared to the primary site. This was attributed to altered cellular immune responses during metastasis, which induced tumor cell senescence. PM tumor cells were surrounded by a substantial number of NK cells, follicular helper T cells (Tfh), and B cells. Additionally, IFN-γ, TNF, and NK regulatory factor IL-15 were also highly expressed in PM tumor cells (Figure 3). Conversely, the primary tumor cells were surrounded by CD4^+^ helper T cells (Th), CD8^+^ cytotoxic T cells (CTL), eosinophils, Th17 cells, and regulatory T cells (Tregs). TNF and IFN-γ at high levels, along with NK cells, provide immune surveillance in PM. Moreover, the expression of angiogenesis-related genes was significantly upregulated in CRC-PM compared to the primary foci. Tfh cells, B cells, and VEGF-A in particular worked together to promote more significant angiogenesis in PM [56].

Another investigation examined immune cell infiltration in the ovarian and peritoneal metastases of CRC. Compared to the original tumor tissue, the infiltrating levels of CD3^+^, CD8^+^, CD20^+^, and CD68^+^ macrophages were noticeably greater. Furthermore, the study demonstrated that the Immunoscore (IS), which is based on the infiltration levels of CD3^+^ and CD8^+^ T cells, and the TBM scores, which are based on the infiltration levels of CD3^+^, CD8^+^, CD20^+^, and CD163^+^ cells in the TME, were effective prognostic indicators for patients with CRC-PM. Patients with high IS (>1), high TBM1 score (≥2), or high TBM2 score (≥2) exhibited significantly longer overall survival (OS), and those with high IS tended to have improved RFS [57].

### 3.2. Phenotypic Abnormality of NK Cells

The PC microenvironment may alter the antitumor activity of NK cells by inducing different impairment characteristics, which in turn promotes tumor evasion from immune surveillance. The phenotypes and functions of tumor-associated NK cells in low-grade and high-grade PC differ, as do their antitumor potential. In peritoneal effusions from low-grade PC, most CD56dim NK cells exhibited a relatively immature NKG2A+KIR-CD57-CD16dim phenotype with significant a downregulation of some activating receptors on the cells, mainly NKp30 and DNAM-1, resulting in dysfunction. In contrast, in peritoneal effusions from high-grade patients, most NK cells were of the CD56dimKIR+ CD57+CD16bright mature phenotype but expressed high levels of PD-1 inhibitory checkpoints and showed more severe defects, such as impaired degranulation. Interestingly, in lower-grade patients with a better prognosis, NK cell phenotypic alterations returned to normal after treatment with CRS/HIPEC. This finding provides clues for developing new immunotherapeutic strategies for PC [58].

The generation of CRC-PM involves a series of immune microenvironmental alterations, which are different from the CRC primary foci greatly, and in-depth studies targeting the local microenvironment of PM will help to reveal the mechanism of PM and open up novel therapeutic approaches.

## 4. Diagnosis and Evaluation

### 4.1. Diagnosis

Due to small metastatic nodules and lack of clinical symptoms, early CRC-PM is often overlooked. Therefore, CRC-PM patients generally wait until they develop severe symptoms such as intestinal obstruction, chronic abdominal pain, and extensive peritoneal effusion to seek medical treatment. Given the diversity and nonspecific nature of CRC-PM symptoms, the staging status at the time of initial diagnosis plays a crucial role in determining patients’ prognosis and quality of life. Thus, early clinical diagnosis and therapy are crucial and demanding. Additionally, a preoperative examination can identify patients who may be entirely resectable.

Currently, the most common diagnostic method for CRC-PM involves combining imaging examinations, such as computed tomography (CT), magnetic resonance imaging (MRI), and positron emission computed tomography (PET-CT), with serum tumor markers like CEA, CA125, and CA19-9. Each examination has its own advantages and disadvantages, allowing for complementarity among these diagnostic techniques. 

#### 4.1.1. Radiological Tests

##### CT

CT is the most widely used imaging test to assess the size and distribution of tumors in clinical practice. The characteristic CT features of PM are linear, banded, soiled, and nodular thickening of the peritoneum, omentum, and mesentery. In more severe instances, there may be cake-like thickening. The administration of contrast agents during the scanning of peritoneal, omentum, and mesangial nodules often results in moderate enhancement. Indirect signs include ascites, thickening, and torsion of the intestinal wall, deformation of the mesenteric vessels, etc. 

The sensitivity and specificity of CT in detecting PM have been reported to be 83% (95% CI, 79–86%) and 86% (95% CI, 82–89%), respectively [59]. The accuracy of CT detection is increasing as equipment and technology improve. For PM, the overall sensitivity, specificity, positive predictive value (PPV), and negative predictive value (NPV) of 64-row CT were 75%, 92%, 90%, and 79%, respectively. However, the average sensitivity of 64-row CT was 89% for lesions 0.5 cm or greater and decreased to 43% for lesions < 0.5 cm [60]. 

The detection rate of CT for small lesions and lesions in special areas, such as diffuse peritoneal involvement, small intestine involvement, and mesenteric involvement, remains low, posing significant challenges in clinical diagnosis and impacting the complete resection of tumors. This low detection rate can be attributed to the low resolution of soft tissues by CT, as well as the similar density of lesions to surrounding soft tissues. Nevertheless, alternative examination methods are still necessary to complement CT and address the limitations in detection rate and avoid clinical oversight.

##### MRI

Compared to CT, MRI has a greater resolution in soft tissues, less kidney toxicity and radiological damage from contrast agents, and greater sensitivity for microscopic PM [61,62]; it is more helpful in distinguishing benign/malignant nodules [63], and it is good at detecting mucinous tumors. In patients with moderate-to-large numbers of ascites, MRI allows the assessment of peritoneal lesions in the visceral wall layer covered by ascites, whereas CT is difficult to assess [64]. PM appears as medium/high signal nodules on fat-suppressed T2-weighted imaging. Although PM smaller than 1 cm is difficult to show on non-contrast MRI, MRI-enhanced scans have good sensitivity to lesions of size < 5 mm [65]. MRI can be used as an evaluation tool for screening patients suitable for CRS/HIPEC therapy, as preoperative MRI evaluation results correlate with intraoperative exploration of intraoperative peritoneal tumor severity, peritoneal debulking thoroughness, and patient survival time [63]. 

In addition, diffusion-weighted imaging (DWI) produces image contrast based on the differences in the mobility of water protons within the tissue and is superior to other imaging techniques in terms of sensitivity and specificity [66]. Even very small lesions (2–3 mm) can show a significantly high signal on DWI-MRI images [67], while ascites and intestinal contents show a low signal. In a Chinese study, the sensitivity, specificity, and accuracy of DWI-MRI in assessing CRC-PM were 80.3%, 84.5%, and 82.1%, respectively [68]. In another study, the sensitivity and specificity of DWI-MRI in detecting peritoneal dissemination of gynecologic malignancies were up to 90.0% and 95.5%, respectively [69]. DWI-MRI has greater value in the preoperative assessment of CRS in patients. When peritoneal carcinoma index (PCI) < 21 was selected as the surgical threshold, the accuracy, sensitivity, specificity, PPV, and NPV of DWI-MRI for selecting patients for resectability were 96%, 100%, 71%, 89%, and 100%, respectively. When PCI < 16 was selected as the surgical threshold, the resectability predictive values were 94%, 100%, 79%, 92%, and 100%, respectively [70]. DWI-MRI still had high diagnostic accuracy when the lesion was located in anatomic locations with low CT detection rates, such as the mesentery, gastrohepatic ligament, and serous membrane surface of the small intestine [71]. 

Despite these advantages, MRI scans are still not the imaging test of choice in most medical institutions. Long scan times, the need for respiratory-gated acquisition for T2WI and DWI, the tendency to produce artifacts, limited recognition of calcifications, and various contraindications have limited the use of MRI [72].

##### PET-CT

PET-CT is widely used for tumor imaging and can be used for diagnosing and evaluating both primary tumors and metastatic lesions due to its wide imaging range. PET-CT has good sensitivity and specificity for PM. A meta-analysis evaluated the diagnostic performance of ^18^F-FDG PET-CT in the detection of PC, which had an overall sensitivity of 0.87 (95% CI 0.77–0.93), a pooled specificity of 0.92 (95% CI 0.89–0.94), and a pooled diagnostic odds ratio (DOR) of 73 (95% CI 34–159) [73]. As for CRC-PM, the sensitivity and specificity of PET-CT for detection were 85% and 88%, respectively [74], so PET-CT can be used as a tool to assess the response to neoadjuvant chemotherapy (NAC) for CRC-PM. 

PET-CT has a few disadvantages, including a high radiation dose, a high price tag, poor spatial resolution, and a limited sensitivity to tumors that are smaller than 4 mm [66]. Moreover, the physiological uptake of ^18^F-FDG by the muscles of the intestinal wall also shows hypermetabolic foci, and the higher SUV results in poorly differentiated tumor imaging in this region [75]. Certain types of CRC, such as signet ring cell carcinoma and mucinous adenocarcinoma, have a low uptake of ^18^F-FDG, resulting in false negatives. ^18^F-FDG PET-CT has a diagnostic accuracy of 91%, a sensitivity of 91%, and a PPV of 100% for the diagnosis of recurrence after the HIPEC treatment of CRC-PM. The mean PCI assessed by PET-CT was 11.4 ± 11.9, slightly lower than surgical PCI (sPCI) (16.6 ± 15.0), and CT was even more severely underestimated PCI (8.4 ± 10.3) [76]. Therefore, despite its limitations in detecting PM, PET-CT is more sensitive and precise for monitoring and following postoperative tumor recurrence compared to CT. Sant et al. compared the accuracy of various imaging modalities in the diagnosis of PM in which CT had pooled sensitivity, specificity, and DOR of 68%, 88%, and 15.9, respectively; DWI-MRI was 92%, 85%, and 63.3, respectively; and PET-CT was 80%, 90%, and 36.5, respectively. Since MRI is more widely used than PET-CT, it looks to be the preferred imaging technology despite its comparable diagnostic performance [66].

Recently, it has been shown that ^68^Ga-DOTA-FAPI-04 PET-CT is more sensitive than ^18^F-FDG PET-CT for diagnosing various types of cancer, especially PC of the stomach. It helps enhance the image contrast and reduce the proportion of missed diagnoses [77], which is also valuable in CRC-PM. 

##### PET-MRI

PET-MRI, the most advanced image fusion device available, has several advantages and applications in the preoperative evaluation and staging of tumors. A study showed that PET-MRI and DWI-MRI evaluations of preoperative CRC patients closely correlated with the sPCI. However, PCI with PET-MRI demonstrated a closer correlation to sPCI than DWI-MRI. Additionally, PET-MRI was found to be more effective in identifying inoperable patients with high tumor load, thereby avoiding unnecessary dissection or laparoscopic surgery [78]. Compared with PET-CT, PET-MRI offers superior soft tissue contrast, which is excellent for the detection of primary tumors and metastases in soft tissues, and provides a more accurate staging of tumors [79]. In detecting CRC-PM, PET-MRI exhibits similar performance to PET-CT but with higher specificity [80]. However, it is worth noting that although PET-MRI is advantageous in monitoring tumor recurrence, its comprehensive use in clinical practice is hindered by the high radiation dose and examination cost associated with the technology.

In addition, PM is often accompanied by the spread of intestinal tumors and decreased intestinal peristalsis. Gastrointestinal contrast examination can observe intestinal peristalsis and contrast agent passage time; determine gastrointestinal motility, intestinal obstruction, and mesenteric contracture; and it is also an important means of determining contraindications for patients with CRC-PM who plan to undergo CRS.

In general, CT is currently the most used imaging method to detect CRC-PM at present, but there are some limitations in diagnosing small nodules. MRI, especially DWI, can assist in the detection of small metastases. PET-CT can be used to monitor recurrence after surgery. Gastrointestinal imaging can also be used as a preoperative evaluation tool for patients scheduled for CRS.

#### 4.1.2. Serum Marker Tests

The detection of serum tumor markers is an important auxiliary judgment and monitoring significance for CRC metastases, and the combined test of CEA, CA125, and CA19-9 is recommended. 

##### CEA

CEA can help determine the degree of tumor invasion, and the probability of CRC patients developing PM with preoperative CEA ≥ 10 ng/mL is significantly higher than that of patients with CEA < 10 ng/mL. T4 tumors, obstructive tumors, and ascites of CEA > 5 ng/dL were associated with cancer-free survival [81]. Ascite CEA levels were also statistically associated with recurrence and peritoneal metastatic recurrence in patients with negative peritoneal cytology [82]. A study by Kozman et al. found that CRC patients with a lower CEA/PCI ratio (<2.3) had a longer median overall survival (mOS, 56 vs. 24 months) and RFS (13 vs. 9 months). The prognostic impact of the CEA/PCI ratio was most pronounced in patients with PCI ≤ 10 (OS, 72 vs. 30 months; RFS, 21 vs. 10 months). An elevated CEA/PCI ratio was an independent poor prognostic factor for both OS and RFS (HR 1.85 and 1.58, respectively). This new assessment method allows both tumor activity and volume to be considered in a single metric and therefore may provide a more accurate indication of tumor biological behavior [83].

##### CA19-9

CA19-9 can assist in determining the proliferative activity of cells in the peritoneal fluid or at the primary site. The elevated CA19-9 concentration is associated with deeper tumor infiltration, PM, and LM. Patients with CA19-9 ≥ 74 U/mL have a 2.9 times higher risk of peritoneal implant metastasis than patients with <74 U/mL. High serum CA19-9 levels and high ascites CEA levels were significantly associated with positive peritoneal wash cytology and peritoneal carcinogenesis, respectively, and were significant predictors [81,84,85]. 

##### CA125

Peritoneal mesothelial cells can release CA125 into the blood upon tumor invasion; therefore, CA125 can assist in determining the extent of peritoneal fluid formation and tumor load, with a significantly higher PPV than other tumor markers. The CA125 concentration correlates with PM and is independent of gender, and the site of the primary tumor does not affect the predictive ability of CA125. Although its sensitivity is lower, the specificity is higher compared to CEA [86]. Research indicates that elevated levels of CEA (>6.5 mg/L) and CA125 (>16 U/mL) are distinct prognostic indicators for patients with CRC-PM who undergo CRS/intraperitoneal chemotherapy (IPC). These markers are not associated with PCI. Importantly, the combination of high CEA and high CA125 significantly increases patient mortality rates by six times [87]. 

It is important to note that not all patients presenting with PM will have elevated serum markers; therefore, serum marker testing can only be used as an adjunctive diagnosis and not as a basis for PM.

#### 4.1.3. Cytological Texts 

Cytological examination of ascitic fluid or peritoneal lavage fluid is currently the gold standard for diagnosing IFCCs in the abdominal cavity. It is routinely recommended as it helps to detect micro-metastases that the naked eye cannot identify. The presence of mucinous ascites contributes to extensive PM [88], and preoperative ascites examination is an important tool to improve the diagnostic rate in this group of patients. The cytology of peritoneal washings at the time of laparoscopic exploration is routinely recommended for patients without extensive preoperative ascites. Positive IFCCs are an independent prognostic factor for poor patient survival [89]. Diagnostic laparoscopic exploration is routinely recommended in patients with high preoperative suspicion of PM, not only to obtain definitive histocytological evidence but also to assess the resectability of PM and guide treatment and prognosis [90].

### 4.2. Evaluation and Staging

Although CRC-PM is already at the advanced stages of the disease, differences in tumor burden and disease severity still have different effects on the prognosis of patients. Reasonable screening of those patients with surgical indications and aggressive surgical procedures to maximize the removal of naked-eye visible tumor tissues have the potential to bring some patients to tumor-free status and achieve clinical cures. Therefore, comprehensive evaluation and accurate grading are the prerequisites to implementing treatment strategies. Despite the various imaging examinations that have been taken, some patients must still undergo a dissection before the diagnosis of PM; their severity can be clarified and the possibility of effective surgical resection can finally be assessed. There are various scoring methods for determining the extent and severity of PM.

#### 4.2.1. PCI 

The PCI is a semi-quantitative indicator of the extent of PM lesions and was proposed by Jacquet and Sugarbaker [91]. Its specific assessment is shown in Figure 4. The PCI is an independent prognostic factor for CRC-PM [92,93]; the higher the PCI score, the worse the prognosis. In addition, the PCI score can also guide treatment decisions. CRS/HIPEC treatment is generally not recommended for patients with a PCI of >20 points [94,95].

#### 4.2.2. Peritoneal Surface Disease Severity Score (PSDSS)

PSDSS is a scoring method based on the clinical symptoms, PCI, and histological characteristics of the primary tumor. Based on the total of the three scores acquired, the PSDSS is divided into four phases: PSDSS I: 2–3 points; PSDSS II: 4–7 points; PSDSS III: 8–10 points; PSDSS IV: >10 points (Table 1). 

PSDSS is an independent prognostic factor that can help predict the survival of patients after CRS/HIPEC treatment. In a retrospective study, the mOS were 45, 19, 8, and 6 months for patients with PSDSS stages I–IV who did not receive CRS/HIPEC and 86, 43, 29, and 28 months for patients who received CRS/HIPEC [96]. These data support the ability of preoperative PSDSS to determine the likelihood of long-term survival of patients after CRS/HIPEC treatment. The PSDSS is relevant in determining the inclusion of patients with CRC-PM in clinical trials and their stratification in clinical trials. 

However, this scoring system does not include biological factors currently known to influence CRC prognosis. Sanchez et al. showed that only the RAS mutation status (HR 2.024; *p* = 0.045) and PSDSS staging (HR 2.90; *p* = 0.009) were independent factors affecting OS of CRC patients treated with CRS/HIPEC. Early PSDSS stage I and II OS was reduced in association with RAS mutations and was not significantly different from PSDSS stage III OS. These results were supported by international multicenter validation. An updated RAS-PSDSS score was proposed by incorporating RAS mutation status. The rubrics were as follows: RAS-PSDSS I: PSDSS score 2–3 + RAS wild type; RAS-PSDSS II: PSDSS score 4–7 + RAS wild type; RAS-PSDSS III: PSDSS 8–10 + RAS wild type or PSDSS score 2–10 + RAS mutation; and RAS-PSDSS IV: PSDSS > 10 + any RAS status. RAS-PSDSS is superior to the PSDSS score alone and provides a rapid and feasible preoperative assessment of expected OS in CRC patients treated with CRS/HIPEC [36].

#### 4.2.3. The Biological Score of CRC-PM (BIOSCOPE) 

Schneider et al. developed a biological score based on the PCI, N stage, G classification, and RAS/RAF status, and this scoring system was divided into four risk groups A~D: BIOSCOPE A (score of 0), which represents no risk factors (PCI < 10, N0, G1–2, RAS/RAF wild type); BIOSCOPE B (score of 1–3); BIOSCOPE C (score of 4–7); and BIOSCOPE D (score of ≥8) (Table 2). The higher the BIOSCOPE score, the worse the patient’s prognosis. Patients in BIOSCOPE A had a median cancer-specific survival (mCSS) of 70 and 65 months in the development cohort and validation cohort, respectively. BIOSCOPE B reflects intermediate risk; the mCSS is 50 months and 39 months, similar to or slightly better than the prognosis of patients receiving CRS/HIPEC reported in the study. The mCSS for BIOSCOPE C patients was 33 and 25 months, respectively, which was significantly better than that of the patients with CRC-PM receiving only systemic targeted chemotherapy. In contrast, BIOSCOPE D patients had the worst prognosis, with an mCSS of only 13 months and 7 months in both cohorts [27]. The new BIOSCOPE score reflects the tumor biology and stratified value of prognosis for patients after CRS/HIPEC treatment. It is a further refinement of the two aforementioned scoring systems, providing better guidance for the assessment of disease status, prognosis determination, and treatment decisions in CRC-PM.

PCI score is extensively utilized and is still the selection criterion for CRS surgery in patients with CRC-PM, which is critical for preoperative evaluation and patient prognosis. It is also crucial that the PCI-based PSDSS and BIOSCOPE incorporate more score indicators.

## 5. Prevention and Treatment

### 5.1. High-Risk Factors and Prevention

Since the early symptoms of CRC-PM are atypical and the role of imaging examination is limited, understanding the risk factors for metastasis can help clinicians anticipate the risk early and formulate intervention plans. The primary goal of PM prevention is to prevent the occurrence of metachronous PM, which is accomplished by taking a series of steps to prevent free cancer cells from spreading to the intraperitoneal cavity during radical CRC surgery or the early postoperative period and killing those that already exist in the peritoneal cavity. 

#### 5.1.1. High-Risk Factors

Patients with colon cancer are more likely to develop synchronous PM than rectal cancer. Right-sided colon cancer, advanced tumors, lymph node stages, and mucinous histopathology are the major high-risk characteristics of PM [97]. Ulcer-infiltrating tumors, obstruction or perforation, increased serum CA125, perineural infiltration, and adjuvant chemotherapy are also high-risk factors for CRC-PM [98,99]. A systematic review showed that the recurrence rate of PM 1 year after the complete resection of synchronous PM was 54–71%, the incidence of PM after the resection of concurrent ovarian isolated metastases was 62–71%, the incidence of PM after the perforation and rupture of the perforated primary tumor was 27%, the incidence of PM after surgery for pT4 tumors was 16%, and the incidence of PM after surgery for mucinous adenocarcinoma or signet ring cell carcinoma was 11–36% [100]. Non-R0 resection of the original site, positive IFCCs, lymph node metastases, insufficient lymph node dissection, etc., are all high-risk variables. 

In addition to patient factors, a further significant cause is iatrogenic factors. During surgical treatment, it is possible that the surgeon might not strictly adhere to the tumor-free principle and squeeze the tumor, causing cancer cells that invaded the serosal layer to fall off or spill into the peritoneal cavity with lymphatic fluid, blood, or intestinal fluid from the severed lymphatic vessels, blood vessels, or intestinal cavity [101], which is also a major cause of postoperative PM. The idea that surgery leads to the rapid progression of PM was first proposed by Sugarbaker et al. in their “tumor cell capture” hypothesis back in 1999 [102]. When the tumor ruptures or is dissected, cancer cells enter the transected lymphatic, blood vessels, and colonize the peritoneal barrier area that has been surgically destroyed [103,104]. The presence of many growth factors and MMPs associated with wound healing in the fibrin deposits on the traumatized peritoneal surface creates a favorable environment for the progression of metastasis [17,105]. Furthermore, increased pressure on the tumor, such as intraperitoneal pressurization during laparoscopic surgery, is a major source of increased exfoliated cells, debris, and embolisms [106].

#### 5.1.2. Prevention

There are a series of preventive measures to help reduce the incidence of PM, including strict adherence to the standard procedure during surgery and the principle of tumor-free treatment. The generation and implantation of IFCCs caused by iatrogenic factors are also supposed to be avoided. For patients with the above-mentioned high-risk factors for PM, prophylactic IPC is recommended. If IFCCs are already present and disseminated, intraperitoneal lavage with distilled water can be considered after specimen removal, which leads to the swelling and rupture of free cancer cells due to hypotonicity, followed by IPC for further removal of IFCCs. Currently, CRS/HIPEC is recommended, the basic principle of which is to kill IFCCs and residual microlesions, and remove active platelets, white blood cells, monocytes, and related cytokines in the peritoneal cavity, reducing the growth of traumatic peritoneal surface tumors. This combination therapy model has shown good results in multiple clinical trials of CRC [8,107,108]. The treatment for CRC-PM will be elaborated on later.

The effectiveness of prophylactic surgery or HIPEC in patients at high risk of developing CRC-PM is somewhat controversial. Prophylactic HIPEC or CRS/HIPEC 1 year after primary tumor resection has been reported to improve disease-free survival (DFS) in CRC patients with high-risk PM factors [109,110], so active early interventions to prevent PM are necessary. The results of the COLOPEC trial, on the other hand, showed that receiving adjuvant HIPEC and systemic chemotherapy after primary tumor resection did not improve the 18-month peritoneal DFS rate compared with the control group (80.9% vs. 76.2%, *p* = 0.28). This suggests that adjuvant HIPEC is not effective in preventing PM [111,112]. The phase 3 study of PHYLOCHIP-PRODIGE 15 also demonstrated that secondary surgery plus HIPEC did not improve the survival of patients at high risk for CRC-PM compared to standard surveillance and that grade 3–4 complication rates were as high as 41% [113]. 

Based on the current evidence, in patients with high-risk CRC-PM, prophylactic measures do not provide better survival benefits, and standard surveillance is sufficient. More clinical studies are needed to further explore the role of intraoperative or early postoperative prophylactic HIPEC therapy.

### 5.2. Treatments

Treatments for CRC-PM are centered on systemic chemotherapy, which may be paired with various regional approaches, including CRS, IPC, and intraperitoneal biotherapy therapies (Figure 5). The combination of CRS and HIPEC may drastically enhance the prognosis of patients with CRC-PM.

#### 5.2.1. Systemic Therapy

##### Chemotherapy

For patients with established CRC-PM, systemic chemotherapy is the basis of treatment. Current first-line chemotherapy regimens for mCRC are FOLFOX, FOLFIRI, CAPEOX, infusion of 5-fluorouracil (5-FU)/leucovorin (LV) or Capecitabine, or FOLFOXIRI with/without combination targeted therapy [114]. This palliative treatment can extend the mOS of patients to 16.3–17.1 months [3]. 

However, SC is less effective in patients with CRC-PM. Compared to metastases from other sites, patients with CRC-PM receiving palliative chemotherapy had a shorter OS (12.7 vs. 17.6 months, *p* < 0.001) and progression-free survival (PFS, 5.8 vs. 7.2 months, *p* = 0.001) [115]. As described previously, CRC-PM is more common in the right-sided and mucinous tumors, which is linked to a worse prognosis for therapy. Furthermore, CRC-PM exhibits a more extreme mesenchymal phenotype and has a higher proportion of stroma components. These factors, along with the presence of a peritoneal–plasma barrier, are less conducive to the effectiveness of cytotoxic drugs and cause poor chemotherapy response. Therefore, a multidisciplinary combination therapy model may appear to be a more effective strategy for CRC-PM. The COMBATAC trial demonstrated that perioperative systemic chemotherapy combined with CRS/HIPEC is not only safe and feasible but also improves PFS in selected CRC-PM patients [116,117]. 

Neoadjuvant or adjuvant chemotherapy is important for patients with CRC-PM. For patients with relatively low mortality and morbidity, the use of NAC before CRS/HIPEC is safe and feasible [118]. The retrospective study of Beal et al. showed that NAC before CRS/HIPEC reduced the PCI scores in patients with CRC-PM and improved mOS (32.7 vs. 22.0 months, *p* = 0.044) [119]. The results of the CAIRO6 trial phase II also demonstrated that perioperative systemic therapy did not increase the incidence of complications (22% vs. 33%, *p* = 0.25), and the objective radiographic and major pathologic response (MPR) rates for neoadjuvant therapy were 28% and 38%, respectively [120,121], demonstrating the safety and efficacy of this treatment modality. The role of perioperative systemic therapy in CRS/HIPEC will be further investigated in the follow-up and the pathological response will be assessed according to KRAS/BRAF status to determine which subgroups may benefit [122,123]. In the systematic evaluation by Flood et al., there were no differences in 1-year and 3-year DFS or 3-year OS, but 5-year OS was higher in the group receiving NAC before CRS/HIPEC [124]. More prospective randomized data are needed to confirm this clinical benefit. In addition, some studies have demonstrated that preoperative OXL-based NAC significantly increases the rate of OXL resistance in patients [125]. So, the value of NAC in terms of survival benefit remains uncertain.

Postoperative adjuvant chemotherapy prevents recurrence and improves prognosis. The observational cohort study of Rovers et al. divided 393 patients with CRC-PM into postoperative adjuvant chemotherapy and close follow-ups, with matching propensity scores showing better OS in the adjuvant chemotherapy group (39.2 vs. 24.8 months, *p* = 0.006) [126]. In another retrospective study, adjuvant chemotherapy had a similar effect, with mOS 8.7 months higher in patients treated with adjuvant therapy, but NAC did not confer a survival benefit in patients with CRC-PM who received CRS/HIPEC [127]. In contrast, the multicenter study by Maillet et al. showed that early postoperative chemotherapy did not improve OS after CRS/HIPEC in patients with CRC-PM compared with the surveillance group [128]. Thus, there are conflicting data on the value of neoadjuvant or adjuvant therapy and no guiding conclusions have been made.

##### Targeted Therapy

Chemotherapy combined with targeted therapy improves survival in patients with CRC-PM compared to chemotherapy alone. The addition of targeted drugs extended mOS by 4.5 months [129]. It is reported that targeted therapy is an independent factor associated with better prognosis in patients with M1c CRC, so this combination therapy mode should be considered the optimal chemotherapy regimen for patients with M1c [130]. 

As mentioned previously, 75% of CRC-PM are CMS4 subtypes with a higher proportion of mesenchymal, and a stronger propensity for hematogenous metastasis so they may benefit from anti-vascular therapy and less from anti-EGFR therapy. An observational cohort study from China showed that median progression-free survival (mPFS 9.6 vs. 6.1 months, *p* < 0.05) and mOS (26.3 vs. 12.7 months, *p* < 0.05) were better in patients with CRC-PM treated with bevacizumab-containing regimens than with cetuximab-containing regimens [131]. 

Pretreatment with bevacizumab resulted in a significant decrease in interstitial fluid pressure in tumor nodules, resulting in deeper penetration of IPC drugs and higher concentrations of intraperitoneal drugs. For patients who cannot have complete CRS, bevacizumab may improve the efficacy of IPC [132]. Although NAC with bevacizumab prior to CRS/HIPEC extends mOS to 27 months in patients with CRC-PM [133], it also increases the incidence of postoperative complications [134], so the potential benefits of using bevacizumab prior to CRS/HIPEC remain to be evaluated. A prospective phase II single-arm trial by Willaert et al. to evaluate the safety and efficacy of perioperative chemotherapy in combination with bevacizumab in patients with PC of colorectal or appendiceal origin undergoing CRS/HIPEC is ongoing [135]. 

Deruxtecan for HER-2-positive mCRC has also been shown to induce objective remission in 45% of patients [136], offering hope for mCRC patients. In an Asian population, the results of the TERRA study also demonstrated that TAS-102 reduced the risk of death in patients with refractory mCRC or intolerant to standard chemotherapy [137]. TAS-102, in combination with bevacizumab, may also improve PFS [138,139]. 

As is known to us all, MMPs play a key role in CRC-PM. After CRC cells colonize the peritoneum, MMPs are significantly upregulated, while the use of the small molecule inhibitor batimastat and specific MMP2/9 inhibitor can prevent fibronectin lysis of CRC cells and reduce the colonization rate [140]. Thus, MMP inhibitors may provide a potential therapeutic strategy for patients with PM. With the application of more effective targeted therapeutic agents, it is possible to further improve the efficacy of peritoneal metastatic cancer.

##### Immunotherapy

In recent years, the effectiveness of immune checkpoint inhibitors (ICIs) in mCRC has been verified successively. The final analysis results of the KEYNOTE-177 study establish the PD-1 ICI pembrolizumab as an effective first-line treatment for patients with MSI-H or deficient mismatch repair (dMMR) mCRC [141]. However, the benefit was limited to patients with dMMR/MSI-H status. Barraud et al. pointed out that the use of ICIs for MSI/dMMR mCRC-derived PC patients can also support an overall response rate of 46% iRECIST, and after a median follow-up of 24.4 months, the mOS and PFS are not reached [142], preliminarily demonstrating the clinical benefit of ICIs in patients with this particular CRC-PM subgroup. 

Since CRC-PM is an end-stage tumor with a distinct biological activity, systemic chemotherapy has a poor effect on it. Adjuvant and neoadjuvant therapy for these patients is still debatable, and additional clinical research is required to determine whether patients can benefit from it. The mechanism of CRC-PM is still unclear, and further exploration of the molecular expression profiles that may predict PM is needed to drive improved targeted, immunotherapeutic strategies.

In conclusion, despite the importance of systemic therapy, its benefit for patients with PC is limited. Depending on the patient’s tumor burden and general condition, local treatment modalities can be considered under multidisciplinary guidance in experienced institutions to further improve the outcome of patients with CRC-PM.

#### 5.2.2. Regional Therapy

##### 5.2.2.1. CRS

CRS refers to the removal of part of the tumor to reduce tumor burden and relieve symptoms when radical tumor removal is not possible, usually in combination with HIPEC. Parikh et al. analyzed 26 included studies and showed that for patients with CRC-PM who received CRS/HIPEC, mOS was 33.6 months, up to 62.7 months (12–63 months), and mDFS was 15 months (9–36 months) [143]. 

CRS involves extensive peritoneal and visceral resection aimed at removing all macroscopically visible tumor lesions and minimizing tumor burden, and it is the cornerstone of this treatment modality [144,145]. CRS improves OS and colorectal cancer-specific survival (CSS) in patients with CRC-PM compared with no CRS (*p* < 0.001) [146]. It not only improves patient prognosis but also the efficacy of CRS alone and CRS/HIPEC are improving, so CRS should be routinely considered for patients with CRC-PM [147]. The peritoneal cavity is carefully explored during the operation, and the severity of the lesion is assessed using the PCI score. The completeness of CRS is one of the most important prognostic factors in patients with CRC-PM [148,149,150,151], assessed using the CC (completeness of cytoreduction) (Table 3) and R score (Table 4) [152]. The CC-0 and CC-1 were considered as complete CRS [153]. An incomplete CRS negatively affects survival [154]; OS in patients with complete CRS is better than incomplete CRS [147,149]. Thus, CC0-1 or R0/1 resection should be strived for in patients with CRC-PM. In a prospective study by Cashin et al., 5-year DFS was defined as the cure, with an overall cure rate of 22%, and a cure rate of 28% in patients with complete CRS, thus suggesting that CRS/HIPEC for large tumors may result in long-term survival or even a cure of the disease and that achieving complete CRS is key [155]. 

Yonemura et al. also suggested that the small intestine and its mesentery involvement are important factors affecting CCR-0 resection. CRC-PM patients with small bowel PCI (SB-PCI) of ≤2 had significantly higher CCR-0 resection rates and survival rates than those of SB-PCI ≥ 3. Therefore, patients diagnosed with SB-PCI ≥ 3 preoperatively should receive NAC or laparoscopic HIPEC therapy to reduce SB-PCI and improve CCR-0 resection rates [150].

The development of near-infrared fluorescence imaging techniques may help improve the thoroughness of CRS. In CRC patients, indocyanine green fluorescence-guided surgery (ICG-FGS) can be used to detect micro-metastases in the peritoneum and other sites such as lymph nodes and liver [156,157]. Unlike ICG, a non-targeted fluorescent contrast agent, the targeted fluorescent contrast agent SGM-101 aggregates in tumor tissues by recognizing CEA expressed in tumor tissues [158]. It is possible to detect preoperative unidentified lesions during surgery, improve PCI, and then affect the resection effect of CRS [159,160]. A recent phase I clinical trial investigated the role of fluorescence imaging using a double-labeled anti-CEA antibody conjugate [^111^In]In-DOTA-labetuzumab-IRDye800CW in guiding CRS in the treatment of CRC-PM, with multimodal imaging that preoperatively detects tumor tissue not recognized by imaging and improves the resection rate of potential lesions during surgery [161]. The development of molecular fluorescence imaging technology provides a feasible way to achieve complete CRS.

Although CRS plays an important role in treating patients with CRC-PM, the resection of microscopic lesions is limited. With local treatment development, CRS combined with intraoperative HIPEC has become the mainstay of treatment for those patients.

##### 5.2.2.2. IPC

The presence of the peritoneal–plasma barrier limits drug absorption into the bloodstream and delays the clearance of the drug from the peritoneal cavity, which can maximize the effect of local treatment while reducing systemic toxicity [162,163]. In addition, the drugs are absorbed through the peritoneum and mesentery and directly converge into the liver through the portal vein system, which has the effect of killing the micrometastases in the liver [164]. Therefore, intraperitoneal local therapy can destroy residual micro-metastases and free cancer cells in the peritoneal cavity, which can be combined with CRS to achieve the best therapeutic effect. IPC modalities include HIPEC, early postoperative intraperitoneal chemotherapy (EPIC), and sequential postoperative intraperitoneal chemotherapy (SPIC). Drugs available for IPC include mitomycin C (MMC), platinum, irinotecan (IRI), 5-FU, gemcitabine, paclitaxel (PTX), docetaxel (DOC), doxorubicin(DOX), raltitrexed (RTX), and melphalan [165]. 

###### HIPEC

HIPEC has become the most widely used IPC modality via the direct cytotoxic effects of hyperthermia, heat-induced TME changes, the synergistic effect of heat and chemotherapy drugs, and the pharmacokinetic advantages of IPC [166]. The intraoperative application of irrigation can make chemotherapeutic drugs and heat evenly distributed in the peritoneal cavity, and the mechanical flushing effect of the irrigation fluid can be used to effectively remove IFCCs and microscopic cancer nodules to prevent peritoneal implantation and metastasis. The most used drugs for HIPEC in patients with CRC-PM are MMC and OXL. Numerous studies have investigated the efficacy of CRS/HIPEC in patients with CRC-PM, as shown in Table 5.

The first randomized controlled trial (RCT) comparing CRS/HIPEC with systemic chemotherapy showed that the mOS of the CRS/HIPEC-MMC group improved by 10 months compared to the systemic chemotherapy group [8]. After a median follow-up of nearly 8 years, the HIPEC group had better mPFS and mCSS than the control group [107]. This Dutch phase III trial demonstrated the benefit of CRS in combination with MMC HIPEC; however, it does not indicate how much of a role CRS and HIPEC, respectively, played in this benefit [176].

Elias et al. recommended HIPEC with a dose of 460 mg/m^2^ of OXL, which allows the peritoneal cavity and tumor to have high concentrations of OXL in the presence of limited systemic absorption [177]. Preliminary results in 24 patients with CRC-PM treated with CRS/HIPEC-OXL showed that 3-year OS was 65% and DFS was 50% [178]. CRS combined with HIPEC-OXL prolonged the mOS of patients with limited resectable PC to approximately 63 months compared to 24 months in the systemic chemotherapy group [168]. The clinical benefit of CRS combined with HIPEC-OXL was demonstrated well, but the results of a recent RCT cast doubt on this benefit.

The French PRODIGE7 study divided patients with CRC-PM into a CRS-alone group and a CRS/HIPEC group. The results of this research showed no significant difference in survival between the control and HIPEC groups, and the incidence of 60-day grade 3 or higher complications was more common in the HIPEC group than in the control group [144]. These results negate the clinical benefits of OXL-based HIPEC therapy. However, there are several limitations to the PRODIGE 7 trial. Ceelen et al. noted that the study may have overestimated the effect of HIPEC in the case of insufficient sample size; second, as a locoregional therapy, peritoneal RFS may be more suitable than OS as the study’s primary endpoint. HIPEC treatment time and perfusate use may also affect the experiment’s results; some patients have been treated with OXL before surgery, which may cause drug resistance [179]. In addition, the investigators did not collect data on RAS or BRAF mutation status, MS status, and the use of systemic chemotherapy regimens, and the effects of HIPEC on these subpopulations and the impact of systemic chemotherapy on the trials were unclear [180]. Further OS subgroup analysis showed that patients with PCI 11–15 could benefit from HIPEC treatment [144], so there may be an additional benefit of adding HIPEC to CRS versus CRS-alone in selected populations. At the same time, the benefits of HIPEC cannot be completely denied due to several limitations. Similarly, Baratti et al. showed that the addition of HIPEC-MMC to CRS did not improve survival in patients with CRC-PM [181]. However, since the study was retrospective and the sample size was small, the benefit of HIPEC could not be completely dismissed. A prospective randomized phase IV clinical trial (GECOP-MMC) is underway to evaluate the efficacy of CRS combined with high-dose MMC-HIPEC in preventing peritoneal recurrence in patients with localized PM (PCI ≤ 20) in colon cancer (Clinicaltrials.gov: NCT05250648) [182]. We expect it to make a difference.

The choice of the optimal HIPEC agent remains a controversial issue. A study by Charrier et al. showed that HIPEC-OXL increased the risk of postoperative bleeding compared with other agents (15.7% vs. 2.6%; *p* = 0.004) [183]. Similar results were found in a retrospective study, in which there was no difference in mOS between the two groups using MMC or OXL, patients treated with HIPEC-OXL had increased complication rates compared to those receiving MMC (66.2% vs. 35.3%; *p* = 0.003), especially bowel weakness, intra-abdominal infections, and urinary tract infections [184]. In contradiction, in the study by Delhorme et al., the DFS and peritoneal DFS (PDFS) were significantly better in the MMC group than the OXL group, and there was no difference in the OS and complication rates between the two groups [185]. Therefore, more prospective RCTs are needed to further elucidate the differences between the two HIPEC regimens.

Other drugs have also been used in HIPEC. RTX also has significant thermal synergies [186]. Qiu et al. validated RTX-based HIPEC inhibited tumor growth in CRC-PM animal models without significant adverse effects [187]. Subsequent research revealed that the combination of RTX and recombinant mutant human TNF-α (rmhTNF) in HIPEC significantly inhibited the formation of tumors [188]. This may provide new therapeutic drug options for those patients. Melphalan has also been proposed as a viable treatment option for patients with peritoneal surface malignancies who are receiving CRS/HIPEC [189], but its pronounced postoperative myelosuppressive effects may limit its use. Furthermore, melphalan does not increase patient survival as much as MMC does [190]. Therefore, using MMC in CRS/HIPEC is still recommended for patients with CRC-PM.

There are significant institutional differences in the current protocols for using HIPEC, and there are no uniform standards for the type, dose, temperature, and duration of the drugs used. The COBOX trial was designed to assess the difference between body surface area (BSA)-based and concentration-based HIPEC-OXL. The pharmacological advantages, complication rates, and mortality rates did not differ between the two groups, whereas receiving concentration-based HIPEC resulted in higher drug concentrations in tumor nodules, higher toxicity, and reduced health-related quality of life (HRQOL) at 3 months postoperatively compared to the BSA group [191]. This difference deserves further exploration in future studies to determine the optimal dose-selection approach.

The effect of HIPEC might be enhanced by some other means. Elevated temperature and prolonged treatment time have been proven to increase the drug absorption rate in HIPEC, promote apoptosis, and inhibit the proliferation of PM tumor cells [192]. Therefore, future trials could target this property to further investigate the effects of HIPEC. The addition of hydrogen peroxide (H_2_O_2_) to the perfusate induces oxidative stress, which can increase the concentration of MMC in the peritoneal cavity, thus enhancing the antitumor effect when used for HIPEC-MMC [193]. García et al. developed a closed abdominal model in which CO_2_ recirculation of perfusate allows a more uniform intra-abdominal temperature and drug distribution [194]. When used in patients with CRC-PM, 49% of the participants achieved a mOS of 3 years [195]. Apart from this, it is reported that high intra-abdominal pressure (IAP) levels of 18–22 mmHg during HIPEC are safe and feasible [196]. High IAP HIPEC after CRS increased cisplatin (CIS) concentrations in the small intestinal mesentery [197]. In the future, these modalities may improve the efficacy of HIPEC in treating patients with PC.

The incidence of adverse events after CRS/HIPEC treatment is as high as 47.7% and the incidence of serious adverse events is 25.6% [198]. Therefore, quantitative prognostic indicators are needed to select patients suitable for CRS/HIPEC. Numerous factors affect the prognosis of patients with CRC-PM treated with CRS/HIPEC. Other than the CC, PCI, and PSDSS score, the status of lymph node involvement (the more lymph nodes involved, the worse the prognosis), rectal primary tumor (PM due to rectal primary have significantly lower survival rates than colonic primaries), the occurrence of perioperative grade III/IV complications, and adjuvant chemotherapy also have an impact on CRS/HIPEC [154]. A PCI of >20 and other poor prognostic factors (lymph node involvement, poor general status after chemotherapy, or progression) are absolute contraindications to CRS/HIPEC [148]. Patients with a PCI of ≤20 who received CRS/HIPEC were significantly better treated than patients with a PCI > 20 [175]. Although the benefit was reduced in patients with a PCI > 20, it was still better than in patients without CRS/HIPEC. On the other hand, if LM is associated with a high PCI, it is a relative contraindication [148]. The results of Elias et al. showed that when the PCI is <12 and LM lesions are <3, the mOS after the complete resection of LM and PM can reach 40 months, and the mOS decreases to 27 months when PCI is ≥12 or LM lesions are ≥3 [199]. Therefore, CRS/HPEC treatment combined with LM lesion resection can be performed in patients with CRC-PM with regional LM. Assaf et al. described the relationship between peritoneal diffusion patterns and survival outcomes in patients with CRC-PM. Scattered peritoneal spread (SPS) was defined as having at least two distant and discrete areas of PCI. Patients in this category have poorer DFS and OS than patients with clustered peritoneal spread (CPS) and are independent of the level of the PCI [200]. Considering this diffusion pattern combined with the PCI may be a prognostic factor for CRS/HIPEC. Hentzen et al. also found that heterochronic CRC-PM was associated with early recurrence after CRS/HIPEC, with a significantly lower mDFS than patients with synchronous CRC-PM (11 vs. 15 months; adjusted HR 1.63, 95% CI 1.18–2.26). Therefore, the time of occurrence of colorectal PM should also be considered [201]. Preoperative PLR (platelet-to-lymphocyte ratio) and low skeletal muscle mass have also been reported as prognostic correlate factors [202,203]. As mentioned earlier, although it is debatable, RAS/RAF and MS status mutation status may affect OS after CRS/HIPEC for CRC-PM and should be considered before surgery.

CRS combined with HIPEC for CRC-PM may improve survival somewhat, but recurrence is inevitable. Disease recurrence occurs in 71.5% of patients who undergo CRS/HIPEC, of which 24.6% are isolated peritoneal recurrences [37]. Repeated CRS/HIPEC in patients with recurrent CRC-PM has been shown in studies to effectively extend the mOS (68 vs. 51 months; *p* = 0.03) and recurrence time (22.0 vs. 10.3 months; *p* = 0.012) compared to a single CRS/HIPEC [204]. Therefore, repeat CRS/HIPEC may be considered for selected patients to prevent PC recurrence. More prospective studies are needed to evaluate the safety and efficacy of repeated CRS/HIPEC.

Although several studies have shown that CRS/HIPEC prolongs survival in patients with CRC-PM (Figure 6), the PRODIGE 7 experiment found that the addition of HIPEC after CRS lacks survival benefit. Further research is needed to explore whether HIPEC after CRS is reasonable. In addition, there is no unified standard for the type of chemotherapy regimen to be used for HIPEC, and how to select the appropriate chemotherapy drug to achieve the best treatment effect remains to be solved. CRS/HIPEC is also not suitable for all patients with CRC-PM and should be considered under multidisciplinary guidance based on factors such as tumor burden, ascites, and strength score. The complication rate after CRS/HIPEC treatment is nearly 50%, and postoperative recurrence is inevitable, which limits its use to a certain extent. Therefore, current guidelines do not recommend CRS/HIPEC as routine treatment. In patients with PM only, complete CRS should be performed, and the addition of HIPEC may be an experimental option that remains to be validated in clinical trials [205].

###### EPIC, SPIC

The disadvantages of HIPEC are that the equipment used in hyperthermia is expensive, and simultaneous HIPEC during surgery prolongs the operation time. Moreover, it is necessary to use cell cycle non-specific drugs with direct cytotoxic and synergistic antitumor activity when combined with hyperthermia. In contrast, EPIC and SPIC are both IPC at room temperature. EPIC is usually administered on the first postoperative day and lasts 4–6 days, while SPIC is continuous treatment for 6 months [209]. Cell cycle-specific drugs such as 5-Fu and PTX are commonly used for EPIC or SPIC.

Albeit the limited benefit of PTX in mCRC [228], the usability of intraperitoneal PTX (IP PTX) in ovarian, gastric, and pancreatic cancers has been confirmed in several studies. Based on this, Murono et al. showed that IP PTX combined with mFOLFOX6/CapeOX plus bevacizumab is safe and feasible in patients with CRC-PM. The outcomes show a 1-year survival rate of 100%, a PFS of 8.8 months (range, 6.8–12 months), and a mOS of 29.3 months, further demonstrating the potential of IP PTX in treating patients with CRC-PM. [229,230]. However, the clinical benefit of IP PTX needs to be validated in further studies because the sample size is too limited to support it in this research.

Compared with CRS alone, in patients with CRC-PM, CRS/EPIC could improve survival [231]. However, Elias et al. found that despite the prolonged operative time, the HIPEC group had lower complication rates, mortality rates, and recurrence rates of PC (26% vs. 56%; *p* = 0.03) than the EPIC group [167]. Another study demonstrated that CRS/SPIC- 5-FU also prolonged the survival of PM patients over systemic chemotherapy [232]. However, in the study by Cashin et al., the mOS and mDFS were better in the intraoperative HIPEC group than in the SPIC group [209]. Despite EPIC and SPIC both being beneficial in patients with CRC-PM, their clinical benefit is inferior to HIPEC. Therefore, HIPEC is still recommended as a first-line treatment strategy for CRC-PM.

Since few data support the superiority of EPIC or SPIC treatment over HIPEC, the combination of postoperative IPC with HIPEC has been performed in several studies. However, CRS/HIPEC plus EPIC is not only not beneficial to patient survival but it also increases the incidence of postoperative complications and prolongs hospitalization, so EPIC is not recommended after CRS and HIPEC [233]. On the contrary, in patients with appendiceal adenocarcinoma, Huang et al. found that the combination of HIPEC and EPIC may provide additional survival benefits for peritoneal spread without increasing postoperative morbidity and mortality [234]. When adding EPIC to CRS/HIPEC, the current evidence is entirely retrospective and conflicting [235], so more studies are needed to further elucidate the role of this combined peritoneal treatment modality.

###### Pressurized Intraperitoneal Aerosol Chemotherapy (PIPAC)

In recent years, PIPAC has been proposed as a new IPC modality, mainly for the treatment of patients with unresectable PC. Unlike other intraperitoneal drug-delivery methods, the drug is sprayed onto the peritoneal surface via pressurized atomization to form an aerosol, which can distribute the drug more evenly. Secondly, the pressure inside the peritoneal cavity is higher during PIPAC treatment, which can increase drug uptake by the tumor tissue [236].

Contraindications to PIPAC include life expectancy of less than 3 months, intestinal obstruction, total parenteral nutrition, decompensated ascites, concurrent tumorous reduction, and gastrointestinal resection, as well as allergy to chemotherapeutic agents. Relative contraindications include extra-abdominal metastases, ECOG score of >2, and portal vein thrombosis [237]. There are no clear indications for PIPAC at present, but it can be used as a prophylaxis for patients with high-risk PM, preoperative tumor shrinkage in CRS, or palliative treatment for patients with diffuse PC [238]. 

PIPAC is available in two intraperitoneal regimens: CIS in combination with DOX or OXL monotherapy, with at least three PIPAC procedures at intervals of 6 ± 2 weeks [237]. The phase I study by Dumont et al. showed that PM patients with gastrointestinal cancer using PIPAC with OXL had a maximum tolerated dose (MTD) of 90 mg/m^2^ [239], while another phase II study recommended 120 mg/m^2^ [240], and this difference may be due to the use of systemic chemotherapy before and between the two PIPAC cycles in the former study [241]. More research is needed to validate the choice of optimal dose.

PIPAC can induce the retraction of PC and improve the survival rate of patients while patients are well tolerated without affecting the quality of life [242,243,244]. A recent systematic review reported that the objective clinical response rate of PIPAC in patients with CRC-derived PC was 71–86%, the mOS was 16 months, and the safety of PIPAC treatment was fine with the main complication rates as follows: intestinal obstruction (0–5%), bleeding (0–4%), and abdominal pain (0–4%) [237]. In patients with unresectable PM, CRS/HIPEC can be achieved by reducing tumor burden with repeated PIPAC. In another study by Alyami et al., 26 patients with PM received CRS/HIPEC after 76 PIPACs (an average of 3), of which 21 (80%) achieved complete CRS and HIPEC [245]. PIPAC, combined with the systemic modality of chemotherapy or targeted therapy, could also be a treatment option for patients with advanced peritoneal disease [246,247,248,249], but its clinical benefit remains to be explored.

The addition of an electrostatic field to PIPAC enhances charged droplet precipitation and tissue penetration and improves the efficacy of the drug. A study by Willaert et al. reported that this approach was safe, feasible, and tolerated well, with 11 of 28 patients experiencing remission after 3 PIPACs [250]. Another research on electrostatic PIPAC (ePIPAC) showed an inadequate histological tumor response compared with standard PIPAC-directed therapy [251]. Therefore, the clinical benefit of ePIPAC needs to be further explored.

The studies on PIPAC are still in the preliminary stage, and the subsequent development will face several challenges, including the standardization of PIPAC technology, drug selection, and protocol development, which are all directions for future research.

##### 5.2.2.3. Intraperitoneal Biotherapy

Biotherapy refers to using biological response modifiers (BRM) to modulate the biological response of cancer patients, thereby inhibiting tumor genesis and development directly or indirectly. The unique microenvironment of the peritoneum provides new ideas for intraperitoneal biotherapy for PM. Common intraperitoneal biotherapeutic strategies, including immunotoxins (ITs), Catumaxomab, immunostimulants, vaccines, cellular therapies, ICIs, and targeted agents, have been validated in preclinical models of CRC-PM. 

###### ITs

The MOC31PE antitoxin consists of a MOC31 monoclonal antibody targeting the epithelial cell adhesion molecule (EpCAM) and Pseudomonas exotoxin A (PE), which can be incorporated into EpCAM-expressing tumor cells and cause cell death by interfering with protein synthesis and inducing apoptosis [252,253]. After CRS/HIPEC, intraperitoneal perfusion of MOC31PE IT can enhance the local inflammatory response of the intraperitoneal cavity and help kill residual tumor cells [254]. With high intraperitoneal drug concentrations and minimal systemic exposure, the ImmunoPeCa phase I clinical trials have shown that intraperitoneal MOC31PE is safe and well-tolerated [255]. After a median follow-up of 34 months, the results show a 3-year OS of 78% and mDFS of 21 months [256]. The safety of intraperitoneal MOC31PE has been illustrated well in patients with CRC-PM undergoing CRS/HIPEC, but the clinical efficacy of intraperitoneal MOC31PE needs to be further confirmed by more large studies.

###### Catumaxomab

Catumaxomab is a trifunctional bispecific antibody that binds to transmembrane glycoprotein-EpCAM on tumor cells and CD3 on T cells and recruits immune helper cells via FcγR binding [257]. Cartumaxomab intraperitoneal injection has been shown to improve median survival in patients with PC of gastrointestinal origin [258]. However, another study demonstrated that intraperitoneal catumaxomab in combination with systemic chemotherapy did not prolong survival in patients with PM from gastric cancer [259]. Follow-up studies are scarce due to more toxicities and fewer indications.

###### Immunostimulant

Lee et al. discovered novel immunotherapies by activating the IFN gene stimulating factor (STING) pathway. The intraperitoneal injection of STING agonists into mice inhibited tumor cell angiogenesis and activated CD8^+^ T cells, which resulted in restraint of peritoneal dissemination of colon cancer, complete elimination of tumors and ascites, and induction of long-lasting anti-tumor immunity [260]. Miller et al. showed that the intraperitoneal administration of Toll-like receptor-9 (TLR-9) agonist CMP-001 activated plasma cell-like dendritic cells (pDC) and triggered the release of IFN-α, enhancing the anti-tumor immune effect, thus improving the survival rate of PC model [261]. In CRC-PM, TLR9 agonists could also significantly enhance the efficacy of the anti-PD-1 treatment, reduce the number of Tim4^+^ peritoneal resident macrophages, and enhance the anti-tumor immunity of CD8^+^ T cells [262]. By enhancing the anti-tumor response, these treatments may provide unexpected benefits to patients with CRC-PM.

###### Vaccines

Oncolytic virus (OV) therapy is a novel immunotherapy that offers new hope for treating malignancies. OV can significantly upregulate PD-L1 in TME, and synergistic effects with PD-L1 inhibitors can induce the regression of PC [263]. Lee et al. found that peritoneal immunotherapy using an oncolytic vaccine virus (JX), which is provided with granulocyte-macrophage colony-stimulating factor (GM-CSF), can reactivate the peritoneal anti-tumor immune response and enhance immune checkpoint blocking. It can effectively inhibit the progression of PC and the formation of malignant ascites. In addition, triple immunotherapy with JX, αPD-1, and αCTLA-4 can produce the most effective anti-tumor immunity and even induce complete tumor regression and long-term OS [264,265]. Apart from this, Tate et al. also demonstrated the feasibility and tolerability of pressurized intraperitoneal nebulized virus therapy (PIPAV) in rats [266], providing a new potential approach for OV therapy to treat PM. Alkayyal et al. also demonstrated that intraperitoneal injection of a vaccine expressing IL-2-expressing Maraba virus cells could significantly slow down the proliferation of PM lesions in mice by inducing the recruitment of activated cytotoxic NK cells into the peritoneal cavity [267]. Preliminary clinical studies have demonstrated that intraperitoneal administration of an oncolytic vaccine virus in patients with advanced PC is well tolerated [268]. The research of tumor vaccines has made great progress in recent years, which may provide new treatment options for patients with CRC-PM in the future.

###### Cell Therapy

Chimeric antigen receptor T-cell (CAR-T) immunotherapy is mainly used in the clinical treatment of patients with malignant hematologic diseases and malignancies. Studies have shown that intraperitoneal CAR-T (IP CAR-T) infusion provides better protection against CEA^+^ peritoneal tumors, which is better than systemic infusion. IP CAR-T infusion associated with immunotherapy, such as depleting antibodies aiming at myeloid-derived suppressor cells (MDSC) and Tregs, can further improve the efficacy against PM [269]. Thus, IP CAR-T may become a new development direction. Moreover, the efficacy of CAR-NK therapy has also been explored. Lin et al. constructed CAR-NK cells targeting NKG2D ligands, and in two patients with PM, intraperitoneal CAR-NK (IP CAR-NK) reduced ascites and significantly decreased the number of tumor cells in samples [270]. IP CAR-NK may be an efficient and safe source of immunotherapy, and both CAR-T and CAR-NK therapy have great potential and broad prospects in CRC-PM patients.

###### ICIs

Kumagai et al. found a significant increase in CD8 ^+^ T cells and a decrease in MDSCs in peritoneal tumors treated with anti-PD-1 monoclonal antibody (aPD-1) in mouse models of gastric cancer-derived PM. Intraperitoneal or intravenous infusion of aPD-1 can both reduce the number of mesenteric metastases by 30–40% [271]. Therefore, it may have the same effect on PCs derived from CRC, which remains to be verified. Si et al. showed that intraperitoneal implantation of a biodegradable implant consisting of DOX and an aPD-1 produced 89.7% tumor suppression in the PM model and the combination of DOX and aPD-1 showed excellent synergistic effects, which is promising for the postoperative treatment of clinical PM [272]. Subsequently, they prepared sustained-release biopolymer immune implants loaded with OXL and resiquimod (R848). Intraperitoneal implantation lasts for 18 days after release, which can enhance anti-tumor immunity and cure 75% of PM cancer mice [273]. Advances in implantable and injectable biomaterials also offer new possibilities for immunotherapy targeting the peritoneal microenvironment.

###### Targeted Therapy

Systemic targeted therapy regimens, such as cetuximab and bevacizumab, have been widely used in the clinic, but effective locally targeted therapies are still under investigation. Jordan et al. showed that intraperitoneal bevacizumab (IP-bev) was well tolerated but poorly controlled for malignant ascites due to gastrointestinal tumors [274]. Further studies are needed to demonstrate the benefit of IP-bev, while the future of intraperitoneal application of other targeted agents is unclear.

Although most intraperitoneal biologic treatments are still in the preclinical or clinical trial stages, they have demonstrated efficacy for CRC-PM and have promising application possibilities in patients with CRC-PM, leading to further investigation.

#### 5.2.3. Radiation and Photodynamic Therapy

##### Radiation Therapy (RT)

For advanced or refractory recurrent malignancies, RT is one of the most effective treatments to alleviate clinical symptoms, prolong patient survival, and improve their quality of life. Patients with metastatic pleural or peritoneal tumors are treated with RT, which can relieve pain and reduce bleeding due to the pleural or peritoneal spread of the primary tumor [275]. Because of the adverse effects of radiation throughout the abdomen, RT is rarely used in the treatment of PC; therefore, it is currently recommended only as a local palliative option for symptom control.

Radioimmunotherapy (RIT) is the internal radiation therapy method that uses radionuclides to label specific antibodies to tumor-related antigens and kills tumor cells via antigen-antibody and radiobiological action. The use of a pretargeted RIT (PRIT), on the other hand, improves clearance and reduces off-target toxicity, and has also been explored in metastatic PC. A study by Chandler et al. showed that the use of anti-GPA33 DOTA-PRIT plus [^177^Lu]LuDOTA-Bn can prolong the survival of CRC-PM mouse models [276]. The study by Rondon et al. also demonstrated that peritoneal PRIT using biological orthogonal chemistry can significantly reduce tumor growth in PM models [277]. Thus, RIT may be an effective treatment for unresectable CRC-PM, as well as an adjuvant or neoadjuvant treatment.

##### Photodynamic Therapy (PDT)

Photodynamic therapy (PDT) is a clinical treatment technique that selectively aggregates on cancerous tissues using a photosensitizer and uses light of specific wavelengths to activate and destroy cancer. PDT can mediate tumor damage by producing reactive oxygen species to directly kill and destroy the tumor-related vascular system and activating the immune system to produce anti-tumor immunity [278]. Gu et al. showed that PDT can modify the microenvironment by recruiting immune cells into the tumor tissue, thereby improving OS in patients with stage III–IV CRC [279]. In CRC-PM, PDT has also been initially explored. According to a systematic review, PDT has some therapeutic benefits in patients with PC following surgery. In animal experiments, PDT has also been shown to considerably reduce tumor growth and increase survival [280]. However, due to the low selectivity of the photosensitizer, PTD can lead to complications such as capillary leak syndrome and intestinal perforation. With the optimization of photosensitizer targeting, CRS combined with PDT showed a survival advantage over CRS alone (52.5 vs. 35.5 days; *p* < 0.005) [281]. As a new therapy, PDT has the advantages of better tissue selectivity, high efficiency, low toxicity, reproducibility, good tolerance, and low side effects, which offers another option for treating patients with PM and has potential clinical applications.

## 6. New Developments

### 6.1. Patient-Derived Tumor Organoid (PDTO)

PDTO can detect the sensitivity to chemotherapeutic agents and predict treatment response and resistance for patients. The clinical benefits of HIPEC remain controversial, and organoid models may provide a pathway to improve HIPEC. Ubink et al. established organoid models from malignant ascites and PM tumors to simulate the response to HIPEC treatment in vitro, and the results showed that MMC was better than OXL, but there was a generalized resistance to single-agent therapy, and, combined with an ATR inhibitor, it could increase the sensitivity of MMC [282]. Forsythe et al. also found differences in HIPEC sensitivity and toxicity based on MMC and OXL under different temperatures and time treatments via the PM organoid model [283]. Organoids can be used as an effective platform for antimicrobial susceptibility testing, which can provide more specific individualized treatment plans for CRC-PM patients to improve treatment precision. 

### 6.2. Liquid Biopsy

As a convenient and non-invasive detection method, liquid biopsy can obtain the genomic information of tumor cells in patients by collecting body fluids, which can be taken multiple times and reflect the changes of tumor heterogeneity in real time, thereby assisting the diagnosis and treatment of tumors, and has certain application value in PC patients. 

On the one hand, the peritoneal cell-free tumor DNA (cfDNA) detected can predict the disease load, thus assessing whether the patient can undergo CRS [284]. On the other hand, patients with positive circulating tumor DNA (ctDNA) in plasma and peritoneal fluid after CRS/HIPEC may have a higher likelihood of peritoneal or systemic relapse [285], so postoperative testing for ctDNA may help predict those at high risk for early recurrence. Erve et al. showed that in 20 patients with isolated CRC-PM patients, tumor-derived cfDNA could be detected in plasma in only 20% of the patients, whereas it could be detected in all peritoneal fluids, and the level of cfDNA in peritoneal fluids was higher than that of plasma. Thus, tumor-derived cfDNA in plasma could be used as a poor biomarker for monitoring CRC-PM, while cfDNA in peritoneal fluids could be used to guide individual sexually based therapy [286]. A recent prospective study showed that peritoneal cfDNA has a sensitivity of 100% and a specificity of 77.3% for the diagnosis of PM and can detect PM in CRC earlier than current radioactive tools [287]. Consequently, the identification of cfDNA in peritoneal lavage fluid alters the clinical practice of monitoring postoperative patients by serving as a novel biomarker and early diagnostic tool for PM.

### 6.3. Drug Delivery Technology

Nanoparticles (NPs) are not easily removed from the peritoneal cavity due to their physicochemical properties, and placing the drug encapsulated in NPs can increase the local drug concentration and prolong the retention time of chemotherapeutic drugs in the peritoneal cavity, thus improving the therapeutic effect.

Studies have shown that tumor growth can be effectively inhibited by constructing a nanoparticle–hydrogel hybrid slow-release drug delivery system loaded with PTX or DOC when injected intraperitoneally into a PC model [288,289]. Poly(cyano-cyanoacrylate) (PACA) NPs loaded with cabazitaxel (CAB) were also shown to improve the therapeutic response in the PM model [290]. Fleten et al. also developed a new delivery system by encapsulating NPs containing CAB-PACA in alginate microspheres and experimentally illuminated that PACAlg not only provides higher peritoneal drug concentrations and tuned drug release rates but also reduces cytotoxicity [291]. Those may become promising therapeutic strategies for CRC-PM patients.

During PIPAC, the spatial drug distribution pattern is not uniform throughout the peritoneal cavity. Göhler et al. recently established hyperthermic intracavitary nano aerosol therapy (HINAT), which can allow for a more uniform distribution of drugs across the peritoneum and deeper drug penetration than with normal PIPAC [292]. In addition to chemotherapy solutions, biomolecules (mRNA, siRNA) are also delivered to the peritoneal cavity by PIPAC nebulization, which can solve the problem of instability of nucleic acid molecules and does not affect transfection efficiency [293,294]. Those open a new area of research for future PC therapy.

Yuan et al. found that PDT mediated by mTHPC@VeC/T-RGD NPs upregulated PD-L1 expression in CRC and sensitized tumors to PD-L1 blockade therapy. The combination of PDT and anti-PD-L1 effectively inhibited the growth of local and distant tumors [295]. Nanosystems represent a new strategy for designing smart PDT therapeutics for more precise and effective treatments, with great potential in inhibiting PM.

Multivesicular liposomes (MVL) are a unique system that allows for sustained release of the drug encapsulated within them, and intraperitoneal injection of OXL-loaded MVL increases OXL uptake and prolongs the duration of OXL. Consequently, it is possible to address the EPIC regimen, which includes redosing, drainage, and circulation from the first postoperative day for five days in sequence [296]. It also has some application value in CRC-PM. 

In summary, developing new drug delivery systems, such as NPs and liposomes, may provide new strategies and improve the therapeutic outcome of patients with PM from CRC.

### 6.4. Radspherin for Short-Range Radiation

A recent study investigated the safety of intraperitoneal administration of ^224^Radium-labeled particles (Radspherin^®^, Oslo, Norway) after 2 days of CRS/HIPEC for treating CRC-PM. Preliminary results showed that Radspherin^®^ was uniformly distributed intraperitoneally, well tolerated at all dose levels, and did not achieve dose-limiting toxicity. There were no serious adverse events associated with Radspherin^®^ as well [297]. Further studies are expected to elucidate this treatment’s safety and clinical benefits.

## 7. Conclusions

### 7.1. Current Status

CRC-PM, one of a group of subtypes with distinctive tumor biological activity, is the outcome of multifactorial, multistage, and multigene interaction. At present, the investigation of the mechanism and molecular biological characteristics of CRC-PM is in the preliminary stage and has not been fully clarified. It is difficult to detect PM in the early stage, and the diagnosis is mainly based on imaging, supplemented by serum marker detection. For patients with high-risk factors, prophylactic IPC is still controversial. The management is still based on systemic treatment, supplemented by local approaches. Although CRS/HIPEC is an effective method for patients with CRC-PM, the clinical benefit of HIPEC remains controversial and still lacks agreed standards. There is also a lack of data from large RCTs of PIPAC as an effective, safe, and well-tolerated emerging treatment. Intraperitoneal biotherapy, RIT, and PDT offer new options, but most are still in preclinical research or clinical trials. 

### 7.2. Future Directions

In the future, more clinical studies and basic experiments should be encouraged to further explore the relevant mechanisms of CRC-PM and whether there are corresponding biomarkers that can predict PM, which will help guide clinicians to better prevent and diagnose PM. In the era of precision medicine, developing efficient and simple diagnostic strategies, early prevention, and individualized treatment to improve the survival rate and quality of life of patients with CRC-PM will be the direction and focus of future research. More large-scale prospective randomized studies are needed to further validate the role of HIPEC and PIPAC and optimize their protocols. Developing new drug delivery systems to improve treatment efficiency and individualized treatment evaluation technologies is also a future development direction.

## Figures and Tables

**Figure 1 cancers-15-05641-f001:**
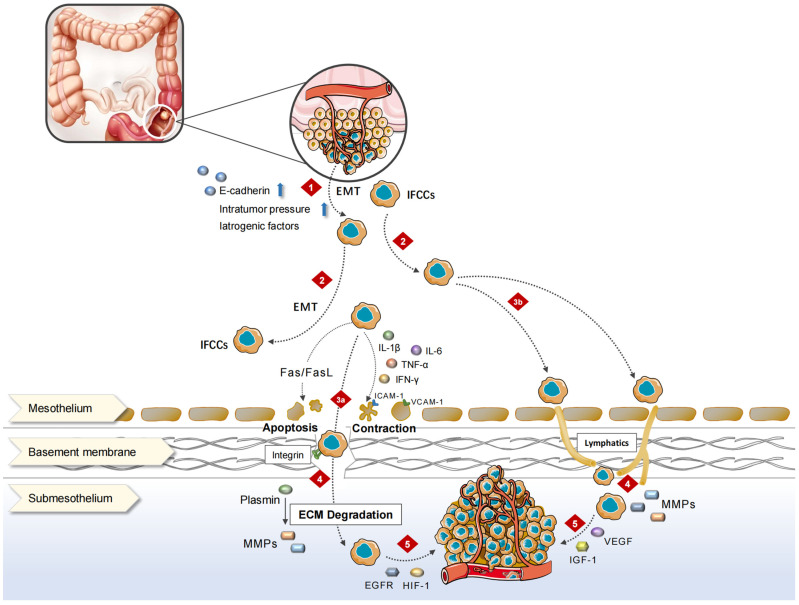
Mechanisms of CRC-PM. 1. In response to a variety of stimuli, proliferating cancer cells break from the original tumor and enter the peritoneal cavity. 2. IFCCs spread throughout the peritoneal cavity, finally implanting on the surface of distant HPMCs. 3. IFCCs cling to the peritoneum via mesothelial or lymphatic pathways. 3a. IFCCs release multiple kinds of pro-inflammatory cytokines that cause mesothelial cell contraction, rounding, or apoptotic shedding, exposing the basement membrane. HPMCs express a wide range of IgCAMs, which bind to glycoproteins on the surface of IFCCs and promote the adhesion of IFCCs to the peritoneum. 3b. IFCCs enter submesothelial lymphatic vessels through openings at the junction of two or more mesothelial cell-lymphatic pores, attaching to the peritoneum. 4. IFCCs invade the subperitoneal region and destroy the ECM via MMPs, causing the peritoneal barrier to rupture. 5. Neovascularization occurs in the tumor-promoting microenvironment created by different metastasis-associated variables, and cancer cells proliferate to form metastatic foci.

**Figure 2 cancers-15-05641-f002:**
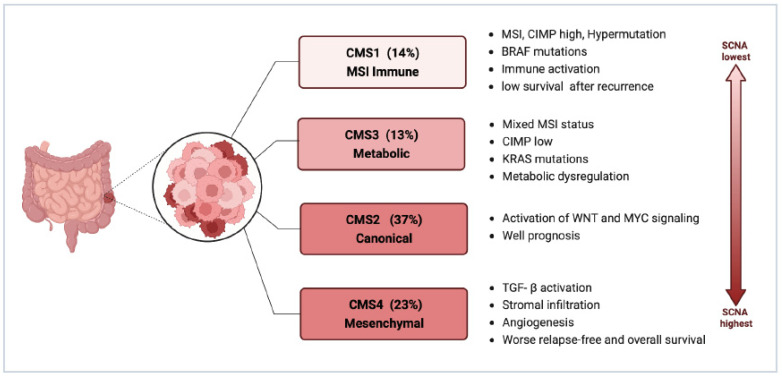
Consensus molecular subtypes of CRC (Created with biorender.com; accessed on 7 November 2023). Abbreviations: CIMP, CpG island methylator phenotype; MSI, microsatellite instability; SCNA, somatic copy number alterations; TGF-β, transforming growth factor-β.

**Figure 3 cancers-15-05641-f003:**
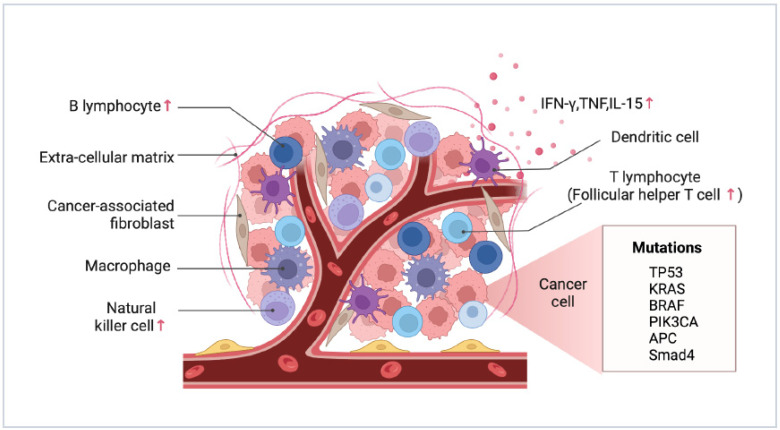
Peritoneal immune microenvironment characterization. The arrows represent cells and factors upregulated in the peritoneal metastasis compared to the primary site (Created with biorender.com; accessed on 7 November 2023).

**Figure 4 cancers-15-05641-f004:**
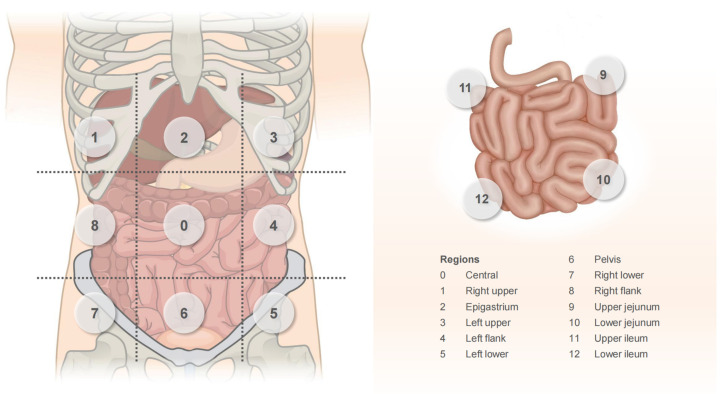
PCI scoring system. The abdominal cavity was separated into 13 sections: central area, central epigastric area, pelvic area, left upper quadrant area, left abdominal area, left lower quadrant area, right upper quadrant area, right abdominal area, right lower abdominal area, and 4 areas of proximal ileum, distal ileum, proximal jejunum, and distal jejunum. The associated lesion size score (LS) is graded on a four-point scale: LS0: no tumor is seen; LS1: tumor ≤ 0.5 cm; LS2: 0.5 cm < tumor ≤ 5 cm; LS3: tumor > 5 cm or fused into clumps. The PCI score is calculated by adding the LS scores for each zone.

**Figure 5 cancers-15-05641-f005:**
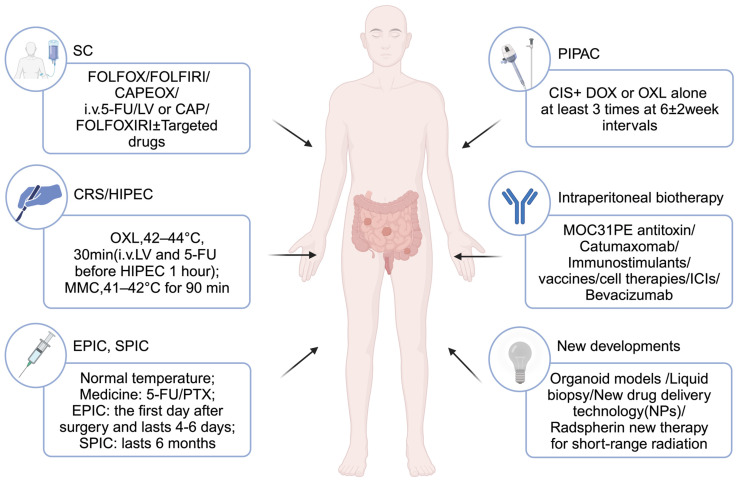
Therapeutic strategies for CRC-PM (Created with biorender.com; accessed on 18 November 2023).

**Figure 6 cancers-15-05641-f006:**
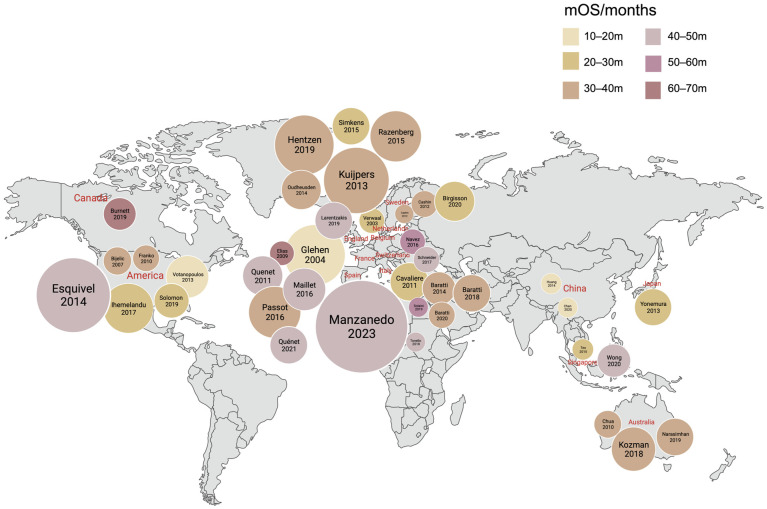
CRS/HIPEC research in various countries [8,83,96,128,144,149,150,151,168,169,170,171,172,173,174,175,181,201,206,207,208,209,210,211,212,213,214,215,216,217,218,219,220,221,222,223,224,225,226,227]. The hue of the nodes corresponds to the duration of mOS for each research. The size of the nodes represents the sample size of each research: the bigger the node, the larger the sample size (Created with biorender.com; accessed on 24 November 2023).

**Table 1 cancers-15-05641-t001:** PSDSS scoring criterion.

Evaluation Indicators	Score
Clinical symptoms	
None	0
Mild Weight loss < 10%, mild abdominal pain, some ascites	1
Severe Weight loss > 10%, persistent abdominal pain, bowel obstruction, symptomatic ascites	6
PCI score	
<10	1
10~20	3
>20	7
Histopathological features	
G1, G2, N−, L−, V−	1
G2, N+ and/or L+ and/or V+	3
G3, Signet ring	9

Abbreviations: PSDSS, peritoneal surface disease severity score; N, lymph node metastases; L, lymphatic vessel invasion; V, vascular invasion.

**Table 2 cancers-15-05641-t002:** BIOSCOPE scoring criteria.

Evaluation Indicators	Score
PCI score	
1–10	0
11–20	2
21–30	4
G stage	
G1	0
G2	0
G3	2
N stage	
N0	0
N1	2
N2	3
RAS/BRAF status	
Wild type	0
NRAS mutation	0
KRAS mutation	1
BRAF mutation	3

Abbreviations: BIOSCOPE, the biological score of colorectal peritoneal metastasis; PCI, peritoneal carcinoma index.

**Table 3 cancers-15-05641-t003:** Completeness of cytoreduction (CC) score.

Grade	Diameter of Residual Tumor
CC0	No residual tumor
CC1	<0.25 cm
CC2	0.25~2.5 cm
CC3	>2.5 cm

**Table 4 cancers-15-05641-t004:** R score.

Grade	Residual Tumor Visible to the Naked Eye	Microscopic Margins
R0	no	negative
R1	no	positive
R2	yes	
R2a	residual lesion < 5 mm	
R2b	5 mm < residual lesion < 2 cm	
R2c	residual lesion > 2 cm	

**Table 5 cancers-15-05641-t005:** Studies of CRS/HIPEC for CRC-PM.

Author, Year, Region	Type	Treatment Arms (Number of Points)	HIPEC Regimen	Overall Survival (Months or Rate)	Morbidity (%)	Mortality (%)	References
Verwaal, 2003/2008, Netherlands	RCT	CRS + HIPEC (n = 54) vs. SC (n = 51)	Open, 41–42 °C, 90 min, MMC 35 mg/m^2^	22.3 m vs. 12.6 m (*p* = 0.032)	NR	8	[8,107]
Elias, 2007, France	Cohort	CRS + HIPEC (n = 23) vs. CRS + EPIC (n = 23)	Open, 43 °C, 30 min, OXL, 460 mg/m^2^	5 y: 54% vs. 28% (*p* = 0.22)	47.8 vs.56.5	0 vs. 8.7	[167]
Elias, 2009, France	Cohort	CRS + HIPEC + SC (n = 48) vs. SC (n = 48)	Open, 43 °C, 30 min, OXL 460 mg/m^2^	62.7 m vs. 23.9 m (*p* < 0.05) 2 y: 81.0% vs. 65.0% (*p* < 0.05) 5 y: 51.0% vs. 13.0% (*p* < 0.05)	NR	NR	[168]
Franko, 2010, America	Cohort	CRS + HIPEC (n = 67) vs. SC (n = 38)	Close, 42 °C, 100 min, MMC 40 mg/3L	34.7 m vs.16.8 m (*p* < 0.001)	NR	NR	[169]
Quenet, 2011, France	Cohort	CRS + HIPEC (OXL + IRI, n = 103 vs. OXL, n = 43)	Open, 43 °C, 30 min, OXL 300–460 ± IRI 200 mg/m^2^	47.0 m vs. 40.83 m (*p* = 0.94)	52.4 vs. 34.9 (*p* = 0.05)	4.9 vs.2.3 (*p* = 0.67)	[170]
Cashin, 2012, Sweden	Cohort	CRS + HIPEC (n = 69) vs. CRS + SPIC (n = 57)	Open, 41–42 °C, 30–90 min, MMC 30/OXL 360–460 ± IRI 360 mg/m^2^	34 m vs. 25 m (*p* = 0.047)	40.6 vs. 29.8 (*p* = 0.02)	4.3 vs. 3.5 (*p* = 0.98)	[171]
Votanopoulos, 2013, America	Cohort	CRS + HIPEC (RC-PM, n = 13 vs. CC-PM, n = 204)	NR	14.6 m vs. 17.3 m 3 y: 28.2% vs. 25.1% (*p* = 0.644)	51 vs. 46 (*p* = 0.78)	5.7 vs. 0 (*p* = 1.0)	[172]
Goéré, 2015, France	Cohort	CRS + HIPEC (n = 139) vs. SC (n = 41)	OXL ± IRI	3 y: 52% vs. 7% (*p* < 0.0001)	44.6 vs. 12.2 (*p* = 0.003)	5.8 vs. 4.9 (*p* > 0.999)	[93]
Navez, 2016, Belgium	Cohort	CRS + HIPEC (n = 52) vs. CRS + HIPEC + LS (n = 25)	Close, 41–42 °C, 30–90 min, MMC 35/OXL 460 mg/m^2^	59.2 m vs. 27.5 m (*p* = 0.06)	15.4 vs. 32.0 (*p* = 0.15)	0 vs. 4.0 (*p =* 0.21)	[173]
Klaver, 2019, Netherlands	RCT	Adjuvant HIPEC+ SC (n = 102) vs. SC (n = 102)	Close/Open, 42–43 °C, 30 min, OXL 460 mg/m^2^	93% vs.94.1% (*p* = 0.82)	14 (HIPEC group)	NR	[112]
Chen, 2020, China	RCT	CRS + HIPEC (n = 14) vs. HIPEC + dCRS (n = 14)	Close, 41–42 °C, 90 min OXL 125/RTX 3/5-FU 175 mg/m^2^	14.5 m vs. 14.3 m (*p* > 0.05)	21.4	NR	[174]
Birgisson, 2020, Sweden	Cohort	CRS + HIPEC (PCI ≤ 20, n = 112 vs. PCI > 20, n = 45) vs. open-close/debulking (n = 44)	Close, 42 °C, 30 min, OXL 460 mg/m^2^	33 m vs. 20 m vs. 9 m (*p* < 0.001)	77 vs. 58 vs. 91 (*p* = 0.08)	1 vs. 4 vs. 30 (*p* < 0.01)	[175]
Beal, 2020, America	Cohort	NAC + CRS + HIPEC (n = 196) vs. CRS + HIPEC (n = 102)	OXL/MMC, 30–120 min	32.7 m vs. 22.0 m (*p* = 0.044)	22.4 vs. 16.7 (*p* = 0.650)	1.5 vs. 2.9 (*p* = 0.411)	[119]
Quénet, 2021, France	RCT	CRS + HIPEC (n = 133) vs. CRS (n = 132)	Close/Open, 43 °C OXL 360–460 mg/m^2^, 30 min	41.7 m vs. 41.2 m (*p* = 0.99)	60 d: 26 vs. 15 (*p* = 0.035)	59 vs. 61	[144]

Abbreviations: CRS, cytoreductive surgery; HIPEC, hyperthermic intraperitoneal chemotherapy; RCT, randomized controlled trial; SC, systemic chemotherapy; MMC, mitomycin C; EPIC, early postoperative intraperitoneal chemotherapy; OXL, oxaliplatin; IRI, irinotecan; NR, not reported; LS, liver sugery; dCRS, delayed cytoreductive surgery; RTX, raltitrexed; 5-FU, 5-fluorouracil.

## Data Availability

The data supporting the conclusion of this review have been included within the article.

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
