# Peer review of "Peritoneal Metastasis: A Dilemma and Challenge in the Treatment of Metastatic Colorectal Cancer"

_cancers, 2023, doi:10.3390/cancers15235641_

Round 1

Reviewer 1 Report

Comments and Suggestions for Authors

thank you for allowing me to review this exhaustive literature on peritoneal carcinosis of colorectal origin. this is a relevant and evolving scientific issue. 

in general, this review of the literature is too long and difficult to read. i suggested that the authors choose from among the topics covered those that seemed most relevant to them. for example, what progress has been made in the last ten years?

as for the results of the literature, tables would be welcome to lighten the manuscript. 

I would also suggest that authors insert one or two concluding sentences at the end of each chapter. For example, it's hard to reach a conclusion on the role of imaging in peritoneal carcinosis.  

Author Response

Thank you very much for providing so much valuable advice. Based on your comments, we have carefully revised the manuscript and figures. Here is a list of point-by-point responses (C: Comment, R: Reply).

C1. In general, this review of the literature is too long and difficult to read. I suggested that the authors choose from among the topics covered those that seemed most relevant to them. for example, what progress has been made in the last ten years?

R1: Thanks for your suggestion. Since our article summarizes the mechanism and process of peritoneal metastasis of colorectal cancer, molecular biological changes, diagnostic methods, existing treatment methods, and representative clinical studies, this inevitably leads to a long article. Our article mainly summarizes some of the latest research advances, for which we have partially cut and condensed, but we have added some missing content considering the comments of other reviewers, so it is still very long. However, we've made adjustments to the structure of the article, which will probably help to understand the article.

C2. as for the results of the literature, tables would be welcome to lighten the manuscript.

R2: Thanks for your comment. We have condensed the content of the corresponding articles presented in Table 3, and we have added Tables 4, 5, and Figure 2 to reduce the burden on the manuscript.

C3. I would also suggest that authors insert one or two concluding sentences at the end of each chapter. For example, it's hard to reach a conclusion on the role of imaging in peritoneal carcinosis.  

R3: Thank you for your suggestion. We have added concluding sentences at the end of each chapter. For example, in the imaging section, we concluded the role that every imaging test plays in patients with CRC-PM.

Reviewer 2 Report

Comments and Suggestions for Authors

Dear Editor;

I have read the manuscript entitled ''Peritoneal metastasis: a dilemma and challenge in the treatment of metastatic colorectal cancer'' prepared for Cancers with great interest.

I have some suggestions to improve this effort,

1- The abstract needs to be more concise and needs to add a conclusion- why the readers need to read this important manuscript

2- Background: can the authors write about why this disease develops with scientific evidence and add the chronological table? and give the subheadings of the chapters in this section. Primarily, can they divide the PM of CRC as synchronous and metachronous!!

3- Pathophysiological process and molecular mechanism of development of Peritoneal metastasis could be the right definition of the chapter: Can the author think more evidence for HER-2 expression and oxidant metabolism as well as more evidence for K-RAS and B-RAF mutations and PIK3 pathway- and MSI status as well as anti-PDL  pathyways- immune check point pathways  again with dividing to subheadings

2- Can author think to write the HIGH RISK AND PREVENTION section to the treatment section - preventive proactive Ip chemo is also started in the management...

The authors give all the tumor micro environment and molecular biology- but still is hard to follow up

They also give CMS subgroups can they add the table and give CMS as the subheading too?

Diagnosis section:

 Can they again subdividing the sections as radiological- cyto-pathologic and laboratory tests - peroperative evaluation - as PSDSS, PCI and other classifications

Management of CRC-PM is summarized with well figure and the explanations

Again can the authors think to write prevention and treatment of the management of CRC-PM with subheadings?

Conclusion: Needs to be with subheadings for the future works as well as clarification what it has been achieved so far- Can the authors think to subdivide to two subheadings again

Finally language editing will make much more readable and  the reference paper in this field.

Comments on the Quality of English Language

Dear Editor;

I have read the manuscript entitled ''Peritoneal metastasis: a dilemma and challenge in the treatment of metastatic colorectal cancer'' prepared for Cancers with great interest.

I have some suggestions to improve this effort,

1- The abstract needs to be more concise and needs to add a conclusion- why the readers need to read this important manuscript

2- Background: can the authors write about why this disease develops with scientific evidence and add the chronological table? and give the subheadings of the chapters in this section. Primarily, can they divide the PM of CRC as synchronous and metachronous!!

3- Pathophysiological process and molecular mechanism of development of Peritoneal metastasis could be the right definition of the chapter: Can the author think of more evidence for HER-2 expression and oxidant metabolism as well as more evidence for K-RAS and B-RAF mutations and PIK3 pathway- and MSI status as well as anti-PDL  pathways- immune checkpoint pathways  again with dividing to subheadings

2- Can the authors think of writing the HIGH RISK AND PREVENTION section to the treatment section - preventive proactive IP chemo is also started in the management...

The authors give all the tumor microenvironment and molecular biology- but still is hard to follow up

They also give CMS subgroups can they add the table and give CMS as the subheading too?

Diagnosis section:

 Can they again subdivide the sections as radiological- cytopathologic and laboratory tests - perioperative evaluation - as PSDSS, PCI, and other classifications

Management of CRC-PM is summarized with good figures and explanations

Again can the authors think to write prevention and treatment of the management of CRC-PM with subheadings?

Conclusion: There needs to be subheadings for future works as well as clarification of what has been achieved so far- Can the authors think of subdividing into two subheadings again

Finally, language editing will make much more readable and the reference paper in this field.

Best regards

Author Response

Thank you very much for providing so much valuable advice. Based on your comments, we have carefully revised the manuscript and figures. Here is a list of point-by-point responses (C: Comment, R: Reply).

C1. The abstract needs to be more concise and needs to add a conclusion- why the readers need to read this important manuscript.

R1: Thanks for your suggestion. We have simplified the abstract and explained the significance of our manuscript.

C2. Background: can the authors write about why this disease develops with scientific evidence and add the chronological table? and give the subheadings of the chapters in this section. Primarily, can they divide the PM of CRC as synchronous and metachronous!!

R2: Thanks for your suggestion. We have divided the background into three parts: epidemiology, occurrence of CRC-PM, and its diagnosis and treatment development. There have not been many milestones in the development of CRC-PM to date, so we are sorry that we have not been able to provide the chronological table. 

C3. Pathophysiological process and molecular mechanism of development of Peritoneal metastasis could be the right definition of the chapter: Can the author think more evidence for HER-2 expression and oxidant metabolism as well as more evidence for K-RAS and B-RAF mutations and PIK3 pathway- and MSI status as well as anti-PDL  pathyways- immune check point pathways again with dividing to subheadings

R3: Thank you very much for your suggestion. Based on this, we have combined pathophysiology and molecular biology into a single chapter. We searched the literature and found additional evidence on K-RAS and B-RAF mutations, MSI status, FBXW7, and lipid metabolism associated with CRC-PM. However, regarding HER-2 expression, PIK3 pathway, anti-PDL pathway, and immune checkpoint pathways, most of what has been reported is their role in metastatic colorectal cancer is not clearly indicated in CRC-PM. Moreover, since our manuscript is too long, we did not include them in our article.

C4. Can author think to write the HIGH RISK AND PREVENTION section to the treatment section - preventive proactive Ip chemo is also started in the management...

R4: Thank you very much for your suggestion.We have adjusted the structure of the article to include prevention in the treatment section. And we moved prophylactic intraperitoneal chemotherapy from the “HIPEC”  to “high-risk factors and prevention”.

C5. The authors give all the tumor micro environment and molecular biology- but still is hard to follow up

They also give CMS subgroups can they add the table and give CMS as the subheading too?

R5: We have added a subtitle to the microenvironment section, which explains the characteristics of the peritoneal immune microenvironment from the differences with the primary lesion and the phenotypic changes of NK cells and removed some less relevant contents. Similarly, we divide molecular biology into high-frequency mutations and CMS. As for CMS typing, we have drawn Figure 3 to represent.

C6. Diagnosis section: Can they again subdividing the sections as radiological- cyto-pathologic and laboratory tests - peroperative evaluation - as PSDSS, PCI and other classifications

R6: Thank you for your suggestion, we have re-divided the diagnosis part into three parts, and as for the perioperative evaluation, each part of the diagnosis is covered, so there is no separate section. At the same time, we found that the fluorescence imaging part overlapped with the CRS part, so we merged and deleted.

C7. Management of CRC-PM is summarized with well figure and the explanations

Again can the authors think to write prevention and treatment of the management of CRC-PM with subheadings?

R7: Thank you for your suggestion. We have fully optimized the structure of the article and added subheadings as much as possible to further clarify the structure of our article.

C8. Conclusion: Needs to be with subheadings for the future works as well as clarification what it has been achieved so far- Can the authors think to subdivide to two subheadings again

R8: Thanks for your suggestion, we have divided the conclusion into two parts: the current status and the future directions.

C9.  Finally language editing will make much more readable and the reference paper in this field.

R9: Thanks for your comments. We have polished our language through a professional organization and hope to make it much more readable.

Reviewer 3 Report

Comments and Suggestions for Authors

The authors present a very interesant manuscript on Peritoneal metastases of colorectal cancer. They have reviewed recent advances in pathofisiology, role of molecular factors, diagnosis and treatment. The work is well written and comprehensively presented.   

The main topic is relevant. During the last years we have registered a revolution in the surgical and medical treatment of Peritoneal Carcinomatosis which has derived in a remarcable improvement in the survival of these patients. A detailed review of the advances in this process, pathogenesis, diagnosis and treatment, is very pertinent.

The information gathered by the authors is specially exhaustive which makes this manuscript more relevant than other similar publications.

The manuscript is well written and presentation is easy to read.

The conclusions are consistent with the evidence presented.

The references are appropiate and provide a great information.

The Tables and Figures are appropiate and informative.

Comments on the Quality of English Language

The english language only needs minor revision

Author Response

Thank you very much for providing  advice for our manuscript. Based on your comments, we have carefully revised the manuscript and figures. Here is a list of point-by-point responses (C: Comment, R: Reply).

C. The authors present a very interesant manuscript on Peritoneal metastases of colorectal cancer. They have reviewed recent advances in pathofisiology, role of molecular factors, diagnosis and treatment. The work is well written and comprehensively presented.   

The main topic is relevant. During the last years we have registered a revolution in the surgical and medical treatment of Peritoneal Carcinomatosis which has derived in a remarcable improvement in the survival of these patients. A detailed review of the advances in this process, pathogenesis, diagnosis and treatment, is very pertinent.

The information gathered by the authors is specially exhaustive which makes this manuscript more relevant than other similar publications.

The manuscript is well written and presentation is easy to read.

The conclusions are consistent with the evidence presented.

The references are appropiate and provide a great information.

The Tables and Figures are appropiate and informative.

The english language only needs minor revision

R: Thank you very much for your comments on our manuscript, as you suggested, we have optimized our language through professional institutions and adjusted the structure of the article with the suggestions of other reviewers.

Reviewer 4 Report

Comments and Suggestions for Authors

In the manuscript, the authors report that they reviewed the mechanism and process of occurrence, molecular biological characteristics, clinical manifestations, and diagnostic methods of CRC-PM. The authors summarized the existing therapeutic means as well as representative clinical studies. The review provides reference for the prevention and control of CRC-PM. But they still need to be improved as mentioned below.

1.      The arrangement of the tables needs to be improved, as shown in Table 2 and 3.

2.      The concepts in the article should be properly and formally described, and some abbreviations should be clearly written as full names, such as SC.

3.      Appropriate figures should be provided for the various characterizations of CRC-PM mentioned in the manuscript.

4.      For patients with established CRC-PM, SC is the basis of treatment. However, SC is less effective in patients with CRC-PM. The reasons for this situation should be discussed.

5.      In the conclusion section, the description of the disadvantages of the CRS combined with HIPEC is too vague.

6.      This manuscript could be improved by changing the structure of the article. The summary of treatment methods related to CRC-PM in this article is too confusing and has not been effectively classified.

Author Response

Thank you very much for providing so much valuable advice. Based on your comments, we have carefully revised the manuscript and figures. Here is a list of point-by-point responses (C: Comment, R: Reply).

C1. The arrangement of the tables needs to be improved, as shown in Table 2 and 3.

R1: Thanks for your comments. Since we added two tables, Table 3 becomes Table 5.

We're sorry we didn't make any changes to these two tables. Would you mind giving us some specific advice? Looking forward to hearing from you.

C2.   The concepts in the article should be properly and formally described, and some abbreviations should be clearly written as full names, such as SC

R2: Thanks for your suggestion. We have replaced SC with systemic chemotherapy, and as for the other acronyms, they take the full name when they first appear.

C3. Appropriate figures should be provided for the various characterizations of CRC-PM mentioned in the manuscript.

R3: Thanks for your suggestion. We drew Figures 2 and 3, respectively on the characterization of the peritoneal metastatic immune microenvironment and the consensus molecular subtypes of colorectal cancer.

C4. For patients with established CRC-PM, SC is the basis of treatment. However, SC is less effective in patients with CRC-PM. The reasons for this situation should be discussed.

R4: Thank you for your suggestion. We have added a discussion of this part of the section on systemic chemotherapy considering some of the features of CRC-PM mentioned above. For example, the high interstitial component of CRC-PM and the presence of a peritoneal plasma barrier can lead to a poor response to chemotherapy.

C5. In the conclusion section, the description of the disadvantages of the CRS combined with HIPEC is too vague.

R5: Thank you for the suggestion, we have perfected this section from the following aspects: the clinical benefits of CRS/HIPEC remain controversial, there is currently no uniform standard,  it can only be performed in selected patients, and complications and postoperative recurrence are inevitable.

C6. This manuscript could be improved by changing the structure of the article. The summary of treatment methods related to CRC-PM in this article is too confusing and has not been effectively classified.

R6: Thank you for the suggestion. We have optimized the structure of the article and refined the content of the article better.

Round 2

Reviewer 1 Report

Comments and Suggestions for Authors

the authors have responded point by point to the questions and comments that have significantly improved the quality of the manuscript. 

Reviewer 2 Report

Comments and Suggestions for Authors

Dear Editor,

The authors made the corrections throughout the whole manuscript.

Their messages are much clearer and they can follow up easily.

Thanks for your effort, I congratulate the authors for their valuable and important paper. 

Comments on the Quality of English Language

Dear Authors,

You made the corrections throughout the whole manuscript.

Your messages are much clearer and it can follow up easily.

Thanks for your effort, I congratulate all the contributors for their valuable and important paper.